# Sterol transfer by atypical cholesterol-binding NPC2 proteins in coral-algal symbiosis

Elizabeth Ann Hambleton[1]*, Victor Arnold Shivas Jones[1], Ira Maegele[1], David Kvaskoff[2], Timo Sachsenheimer[2], Annika Guse[1]*

[1]Centre for Organismal Studies (COS), Universität Heidelberg, Heidelberg, Germany; [2]Heidelberg University Biochemistry Center (BZH), Universität Heidelberg, Heidelberg, Germany

**Abstract** Reef-building corals depend on intracellular dinoflagellate symbionts that provide nutrients. Besides sugars, the transfer of sterols is essential for corals and other sterol-auxotrophic cnidarians. Sterols are important cell components, and variants of the conserved Niemann-Pick Type C2 (NPC2) sterol transporter are vastly up-regulated in symbiotic cnidarians. Types and proportions of transferred sterols and the mechanism of their transfer, however, remain unknown. Using different pairings of symbiont strains with lines of *Aiptasia* anemones or *Acropora* corals, we observe both symbiont- and host-driven patterns of sterol transfer, revealing plasticity of sterol use and functional substitution. We propose that sterol transfer is mediated by the symbiosis-specific, non-canonical NPC2 proteins, which gradually accumulate in the symbiosome. Our data suggest that non-canonical NPCs are adapted to the symbiosome environment, including low pH, and play an important role in allowing corals to dominate nutrient-poor shallow tropical seas worldwide.
DOI: https://doi.org/10.7554/eLife.43923.001

*For correspondence:
liz.hambleton@cos.uni-heidelberg.de (EAH);
annika.guse@cos.uni-heidelberg.de (AG)

Competing interests: The authors declare that no competing interests exist.

## Introduction

Many plants and animals cultivate symbioses with microorganisms for nutrient exchange. Cnidarians, such as reef-building corals and anemones, establish an ecologically critical endosymbiosis with photosynthetic dinoflagellate algae (*Douglas, 2010*) (family *Symbiodiniaceae*) (*LaJeunesse et al., 2018*). Their symbionts reside within endo/lysosomal-like organelles, termed symbiosomes, and transfer photosynthetic products to their hosts (*Muscatine, 1990*; *Yellowlees et al., 2008*). In addition to sugars that mostly provide energy, recent studies hint at the importance of the transfer of various lipids including sterols (*Crossland et al., 1980*; *Battey and Patton, 1984*; *Revel et al., 2016*). Sterols are essential building blocks for the cell membrane and endomembrane systems, in the form of cholesterol and other sterol variants. Cnidarians are sterol auxotrophs (*Baumgarten et al., 2015*; *Gold et al., 2016*) that must acquire these essential compounds from diet and/or symbionts (*Goad, 1981*). In line with this, non-canonical variants of the conserved cholesterol transporter Niemann-Pick Type C2 (NPC2) are among the most up-regulated genes in symbiotic *Exaiptasia pallida* (commonly *Aiptasia*) and *Anemonia viridis* anemones (*Dani et al., 2014*; *Lehnert et al., 2014*; *Kuo et al., 2010*; *Ganot et al., 2011*; *Wolfowicz et al., 2016*). Dinoflagellates synthesize various sterols, many of which are found in symbiotic cnidarians (*Bohlin et al., 1981*; *Withers et al., 1982*; *Ciereszko, 1989*); however, the specific combinations of transferred sterols, as well as the mechanism of this transfer remain unknown. To what extent is the specific mix of transferred sterols controlled by the host, symbiont, or both – reflecting physiological relevance – and how is such selective transport achieved?

**eLife digest** Coral reefs are the most biodiverse marine ecosystems on our planet. Their immense productivity is driven by friendly relationships, or symbioses, between microbes called algae and the corals. Related organisms, such as anemones, also rely on these close associations. The algae use energy from sunlight to make sugars, cholesterol and other molecules that they supply to their host. In exchange, the host's cells provide homes for the algae inside specialist, acidic structures called symbiosomes.

Corals and anemones particularly need cholesterol and other 'sterol' molecules from the algae, because they are unable to create these building blocks themselves. In mammals, a protein known as Niemann-Pick Type C2 (NPC2) transports cholesterol out of storage structures into the main body of the cell. Corals and anemones have many different, 'atypical' NPC2 proteins: some are produced more during symbiosis, and these are mainly found in symbiosomes. However, it was not known what role these NPC2 proteins play during symbioses.

Here, Hambleton et al. studied the symbioses that the anemone *Aiptasia* and the coral *Acropora* create with different strains of *Symbiodiniaceae* algae. The experiments found that the strain of algae dictated the mixture of sterols inside their hosts. The hosts could flexibly use different mixes of sterols and even replace cholesterol with other types of sterols produced by the algae. Atypical NPC2 proteins accumulated over time within the symbiosome and directly bound to cholesterol and various sterols the way other NPC2 proteins normally do. Further experiments suggest that, compared to other NPC2s, atypical NPC2 proteins may be better adapted to the acidic conditions in the symbiosome. Taken together, Hambleton et al. propose that atypical NPC2 proteins may play an important role in allowing corals to thrive in environments poor in nutrients.

The first coral reefs emerged over 200 million years ago, when the Earth still only had one continent. Having built-in algae that provide the organisms with nutrients is thought to be the main driver for the formation of coral reefs and the explosion of diversity in coral species. Yet these ancient relationships are now under threat all around the world: environmental stress is causing the algae to be expelled from the corals, leading to the reefs 'bleaching' and starving. The more is known about the details of the symbiosis, the more we can understand how corals have evolved, and how we could help them survive the crisis that they are currently facing.

DOI: https://doi.org/10.7554/eLife.43923.002

## Results and discussion

To answer these questions, we took advantage of the availability of distinct strains of *Symbiodiniaceae* symbionts with different and complex sterol compositions (*Bohlin et al., 1981*; *Withers et al., 1982*; *Ciereszko, 1989*), and of various hosts. Besides the coral *Acropora digitifera*, we investigated different host lines of the symbiotic anemone *Aiptasia*, an emerging model system for coral-algal symbiosis (*Tolleter et al., 2013*; *Neubauer et al., 2017*). We used gas chromatography/mass spectrometry (GC/MS) to semi-quantitatively profile relative sterol abundances in three compatible symbiont strains (*Xiang et al., 2013*; *Hambleton et al., 2014*), and related this to sterol abundances in the coral and in three *Aiptasia* laboratory lines (*Grawunder et al., 2015*), with or without symbionts (*Figure 1*, *Figure 1—source data 1*). First, to validate our assay and to show that algal sterols are indeed transferred to host tissue, we determined the host sterol composition without symbionts (aposymbiotic), in symbiosis with recent dietary input (two weeks since last feeding, 'intermediate'), and in symbiosis with essentially no dietary input (five weeks since last feeding, 'symbiotic'). For the *Aiptasia* F003 host line, this revealed a gradual transition from an initial aposymbiotic, food-derived cholesterol profile to a cholesterol-reduced, algal sterol-enriched symbiotic profile that was also found in the symbiont-free eggs (and is thus present in host tissue) (*Figure 1A*). We also compared the sterol composition of coral symbiotic polyps collected from the wild to that of their symbiont-free eggs, which again proved nearly identical sterol compositions (*Figure 1A*) and unambiguously revealed symbiont-to-host tissue transfer. Taken together, this suggests that symbiont-derived sterols can functionally replace dietary cholesterol without any further chemical conversion by the host. Moreover, the sterol content of the hosts is highly plastic, and sterols are used flexibly as they become available from food and/or symbionts.

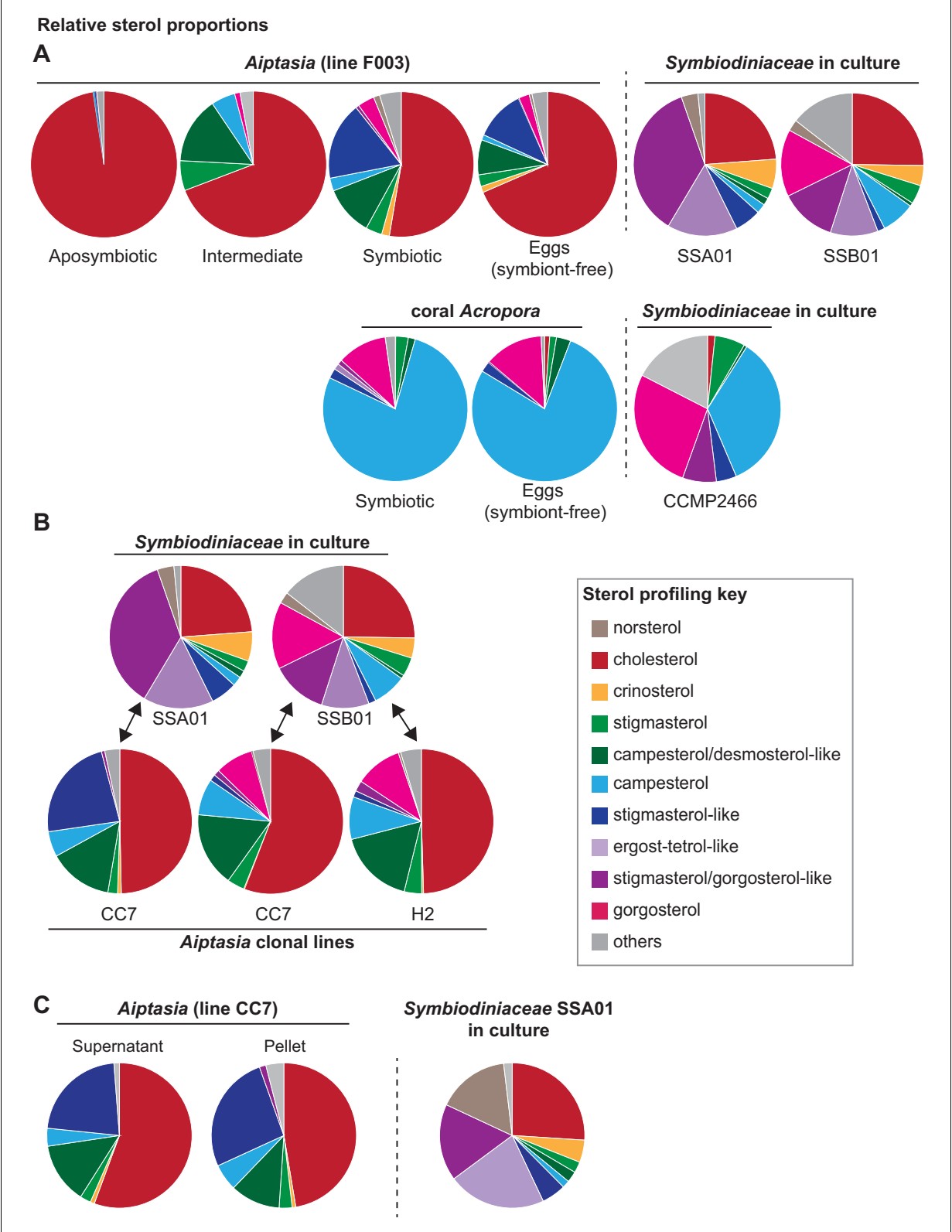

**Figure 1.** Transfer of symbiont-produced sterols reflects control by both host and symbiont. (**A**) Gas chromatography/mass spectrometry (GC/MS)-generated sterol profiles of the given organisms, with relative composition (%) of each sterol in key. Values, *Figure 1—source data 1*. Symbiont-free animals ('aposymbiotic') were fed *Artemia* brine shrimp comprising nearly only cholesterol (*Tolosa et al., 2011*). 'Intermediate' were symbiotic *Aiptasia* more recently starved of brine shrimp diet than 'symbiotic' animals. *Aiptasia* strain F003 hosts *Symbiodiniaceae* strains SSA01 and SSB01

*Figure 1 continued on next page*

*Figure 1 continued*

(*Grawunder et al., 2015*). *Acropora digitifera* endogenous *Symbiodiniaceae* are uncultured but closely related to the cultured strain CCMP2466 (see Materials and methods). (B) GC/MS-generated sterol profiles of *Symbiodiniaceae* strains in culture (upper row) and in symbiosis with adults of different *Aiptasia* host lines (*Grawunder et al., 2015*) (lower row). (C) Sterol profiles of *Aiptasia* CC7 homogenate separated by centrifugation into symbiont-enriched (Pellet) and symbiont-depleted (Supernatant) fractions with the corresponding *Symbiodiniaceae* strain SSA01 in culture.

DOI: https://doi.org/10.7554/eLife.43923.003

The following source data and figure supplements are available for figure 1:

**Source data 1.** Relative sterol compositions of samples in pie graphs.

DOI: https://doi.org/10.7554/eLife.43923.006

**Figure supplement 1.** Some of the symbiont-produced sterols.

DOI: https://doi.org/10.7554/eLife.43923.004

**Figure supplement 2.** GC/MS-generated sterol profiling of symbionts in culture *vs.* in symbiosis.

DOI: https://doi.org/10.7554/eLife.43923.005

We next focused on the sterol compositions in different symbiont-host pairings, to determine how these would change upon switching of either symbiont or host line. To this end, we investigated the same *Aiptasia* line CC7 hosting distinct symbionts (SSA01 or SSB01, see Materials and methods) with different symbiont profiles; and the same symbiont (SSB01) in two distinct host lines (CC7 and H2), as well the symbiont CCMP2466 similar to that in *Acropora*. We found that *Aiptasia* CC7 hosting *Symbiodiniaceae* strain SSA01 contained a large proportion of stigmasterol-like sterol (dark blue, *Figure 1B*) when compared to campesterol (light blue, *Figure 1B*). In contrast, the same *Aiptasia* line hosting strain SSB01 contained minimal stigmasterol-like derivatives compared to campesterol, as well as the unique sterol gorgosterol (light blue and pink, respectively, *Figure 1B*), characterized by an unusual cyclopropyl group (*Ciereszko, 1989*). (*Figure 1—figure supplement 1*). A very similar sterol profile was observed when the same symbiont (SSB01) infected the H2 host line, indicating that the host sterol profile was largely symbiont-driven. Likewise, in *Aiptasia* line F003 hosting both SSA01 and SSB01, the sterol proportions reflect both symbionts: a dominance of stigmasterol-like sterol (reflecting SSA01) together with gorgosterol (reflecting SSB01) (*Figure 1A*). We also compared the sterol profile of *Acropora* colonies collected from the wild to that of a closely related symbiont CCMP2466 in laboratory culture and found a strong enrichment for gorgosterol and campesterol at the expense of stigmasterol-like sterols – highly reminiscent of the trend previously observed in the SSB01/CC7 and SSB01/H2 pairings (*Figure 1A*). We thus observed two major patterns of sterol transfer in our symbiont-host combinations – one enriching for stigmasterol-like sterols (combinations SSA01/CC7 and SSA01 +SSB01/F003), and another one enriching for gorgosterol and campesterol (combinations SSB01/CC7; SSB01/H2; and CCMP2466/*Acropora)*. This suggests that selective sterol transfer and/or accumulation by the host may occur.

Moreover, symbionts may change their sterol synthesis profile as symbiotic *vs.* free-living cells. To address this, we separated anemone homogenates by centrifugation into symbiont-enriched (pellet, although substantial host tissue remained, *Figure 1—figure supplement 2A* and *Figure 1—source data 1*) and symbiont-depleted (supernatant) fractions, for which the sterol profiles could be directly compared to free-living symbionts cultured under similar conditions. This revealed that certain sterols were absent in symbiont-enriched pellets yet present in symbiont cultures (*Figure 1C*, *Figure 1—figure supplement 2B*). For example, stigmasterol/gorgosterol-like (dark purple) and ergost-tetrol-like sterol (light purple) are proportionally highly abundant in cultured symbionts, yet are basically absent in all pellet samples (*Figure 1C*, *Figure 1—figure supplement 2B*). This suggests that synthesis of at least some sterols changes in residence *vs.* in culture, providing further support that the symbiont has a major influence on which specific composition and proportion of the sterols are transferred during symbiosis. Further, cultured symbionts exhibited some degree of plasticity of sterol profiles under various culturing conditions (e.g. SSA01 in *Figure 1A vs. Figure 1C*).

To elucidate possible molecular mechanisms how symbiont-hosting cells may influence sterol transfer from the symbiont, we focused on non-canonical members of the highly conserved NPC2 protein family (*Dani et al., 2014*; *Lehnert et al., 2014*). The current hypothesis in the field is that non-canonical NPC2s may specifically facilitate transfer of symbiont-produced sterols in cnidarian-algal symbiosis (*Revel et al., 2016*; *Baumgarten et al., 2015*; *Wolfowicz et al., 2016*; *Dani et al., 2017*). However, NPC2s may serve other purposes, for example signaling (*Baumgarten et al., 2015*;

*Dani et al., 2017*), and mechanistic analyses of NPC2 function are lacking. To characterize them further, we first compared the genomic complement of NPC2 homologues in symbiotic cnidarians to that of non-symbiotic metazoans, uncovering several previously unidentified homologues in the reef-building corals and other taxa (asterisks, *Figure 2A*, *Supplementary file 1*). A Bayesian tree reconstruction placed all canonical NPC2 family members (identified by three shared introns) on a large multifurcation, and all previously and newly identified non-canonical NPC2 (identified by the absence of introns due to retrotransposition [*Dani et al., 2014*]) to a basal position, most likely attracted by the *Capsaspora* outgroup NPC2s. This indicated higher sequence divergence of non-canonical NPC2s; and in line with this, they contain only around half as many residues under negative (purifying) selection (35 to 61) as canonical NPC2s and twice as many residues under positive (diversifying) selection (12 to 5) (*Figure 2B*). Our analysis also revealed that non-canonical NPC2 homologues are confined to cnidarians within the anthozoan class, as they did not appear in the earlier-branching sponge *Amphimedon* nor in the hydrozoans *Hydra magnipapillata* and *Hydractinia echinata*. Notably, the occurrence of non-canonical NPC2s appeared to correlate with symbiotic state: the symbiotic anthozoans (*Aiptasia*, *Acropora, Montastrea*) have several non-canonical NPC2 homologues (3, 3, and 2, respectively). In contrast, the non-symbiotic anemone *Nematostella* displays evolutionary traces of a single non-canonical NPC2, which either failed to expand or underwent higher loss (*Figure 2A*).

We next investigated the expression of all *Aiptasia* NPC2s in vitro and in vivo (*Figure 3*). As determined by qPCR and Western blotting using custom-made antibodies (*Figure 3—figure supplement 1*), two of the three non-canonical NPC2 homologues displayed substantially higher expression at the transcript and protein levels in symbiotic but not aposymbiotic animals (closed blue symbols; *Figure 3A+B*). The third non-canonical NPC2 homologue was highly expressed in both symbiotic and aposymbiotic animals, yet more so in symbiotic animals. Conversely, canonical NPC2s were highly expressed in both symbiotic and aposymbiotic animals (closed red symbols). Likewise, the non-symbiotic anemone *Nematostella* exhibited ubiquitously high expression of canonical NPC2 genes (open red symbols), whereas the non-canonical NPC2 gene was highly expressed only upon feeding (open blue symbols). Aposymbiotic embryos of the symbiotic coral *Acropora*, as well as *Nematostella* embryos, contained maternally provided canonical NPC2 transcripts, suggesting that these are required for development (*Figure 3—figure supplement 2*). Notably, several canonical NPC2s in *Aiptasia* (XM_021046710, XM_021041174) and *Nematostella* (XM_001635452) may be 'in transition' to becoming non-canonical: they were expressed at intermediate abundances between the two groups, and they responded to symbiosis (*Aiptasia*) or feeding (*Nematostella*) (red square and triangles, *Figure 3A + 3B*). Some of their intron/exon structures reflected those of the non-canonical group (red triangles, *Figure 2A*). Immunofluorescence analysis revealed that the non-canonical NPC2s decorated intracellular symbionts in *Aiptasia* in vivo (*Figure 3C + D*), consistent with previous data in *Anemonia viridis* (*Dani et al., 2014*; *Dani et al., 2017*). The NPC2 signal appears to be restricted to the symbiosome and absent from the cytoplasm of the symbiont-containing cell (*Figure 3—figure supplement 3*). We noted that non-canonical NPC2s decorate some but not all symbionts (*Dani et al., 2014*; *Dani et al., 2017*), suggesting that at any given time, symbiosomes are a dynamic group of specialized organelles. To gain further insight into the NPC2-decorated symbiosome dynamics, we measured the spatio-temporal regulation of non-canonical NPC2s in *Aiptasia* larvae establishing symbiosis ('infection') with *Symbiodiniaceae* strain SSB01 (*Hambleton et al., 2014*; *Bucher et al., 2016*). Indeed, non-canonical NPC2 slowly decorated intracellular symbionts over time (*Figure 3E*, *Figure 3—figure supplement 4*). This localization ranged from weak 'grainy' patterns to stronger 'halos' around symbionts (arrows, *Figure 3C*). We quantified infection rates, symbiont load of individual larvae, and non-canonical NPC2 signal intensity (*Figure 3F*, *Figure 3—figure supplement 5*). We found that infection rates remained steady after removal of symbionts from the environment, whereas the proportion of larvae showing non-canonical NPC2 signal continued to increase to eventually include the majority of infected larvae (*Figure 3F*). Concordantly, the proportion of symbionts within each larva surrounded by NPC2 signal also increased over time, as did the signal strength (*Figure 3F*). Finally, infected larvae displaying any NPC2 signal generally contained a higher symbiont load than their infected, unlabelled counterparts (*Figure 3—figure supplement 5*). Thus, non-canonical NPC2 is increasingly expressed and recruited to symbionts over time, suggesting that non-canonical NPC2 function becomes important primarily once symbiosomes become 'mature'.

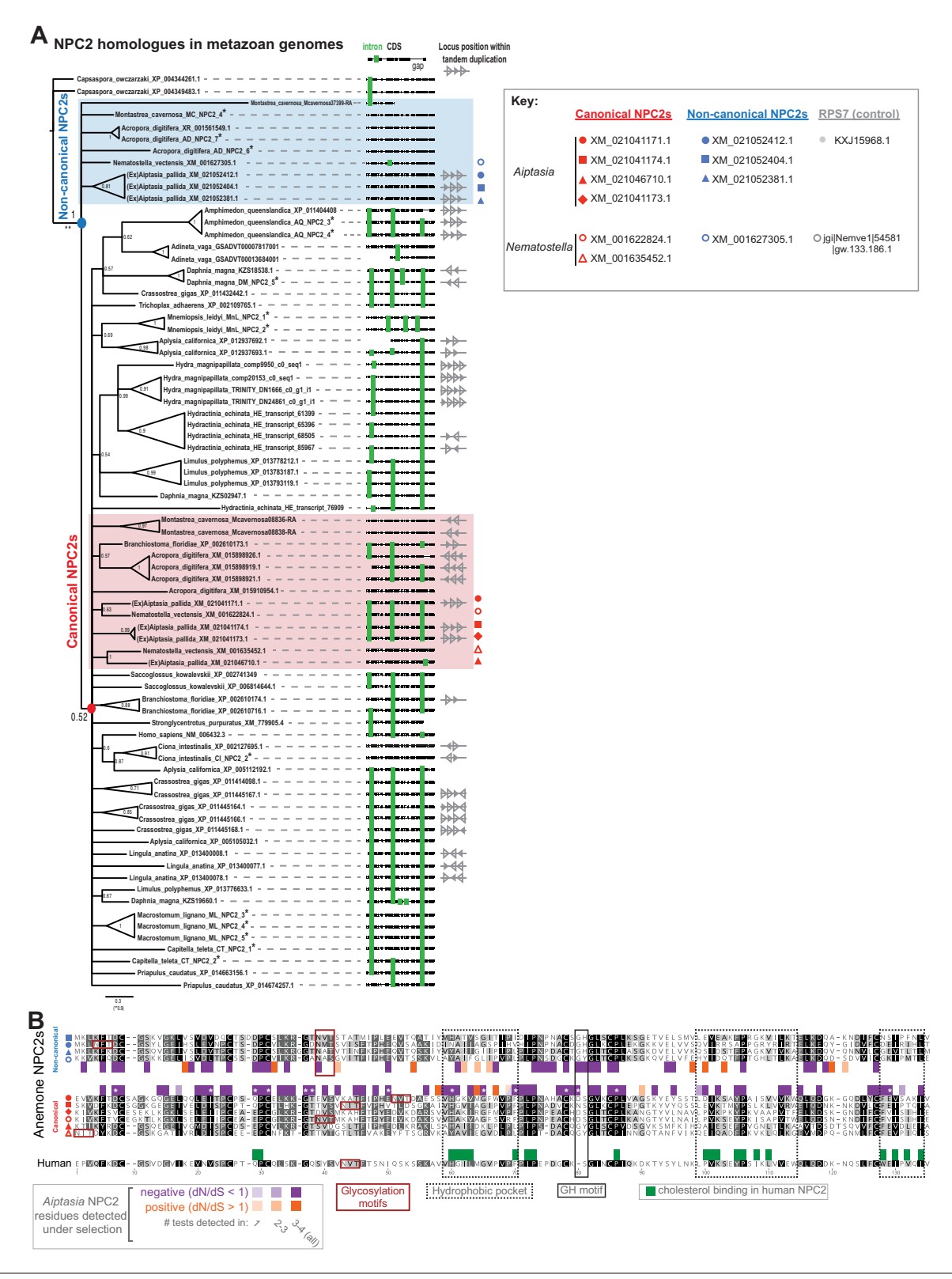

**Figure 2.** Symbiotic anthozoans have expanded NPC2s with characteristics of differential adaptive evolution. (**A**) Consensus Bayesian phylogeny of NPC2 homologues in metazoan genomes with anthozoan non-canonical (blue shading) and canonical (red shading) homologues. Also shown are alignments with intron/exon boundaries (green bars) and tandem duplication of NPC2 loci (where genome assemblies allow). Node values, posterior probabilities. Asterisks, new homologues from this study (***Supplementary file 1***). (**B**) Alignment of anemone and human NPC2 proteins, with shading by

*Figure 2 continued on next page*

*Figure 2 continued*

conservation. Shown are residues under positive (orange) or negative (purple) selection per NPC2 group as found in multiple tests of non-synonymous/synonymous substitution rates (dN/dS) in HyPhy (*Pond et al., 2005*); asterisks, significant in all tests. Indicated are also several functional regions in human NPC (*Xu et al., 2007*; *Friedland et al., 2003*; *Ko et al., 2003*; *Wang et al., 2010*; *McCauliff et al., 2015*).

DOI: https://doi.org/10.7554/eLife.43923.007

As a first step towards elucidating NPC2 function during symbiosis, we investigated the effect of global sterol transport inhibition by treating symbiotic and aposymbiotic adult *Aiptasia* with the drug U18666A, a competitive inhibitor of the NPC2 binding partner NPC1 that is required for efficient cholesterol egress from lysosomes (*Liscum and Faust, 1989*; *Cenedella, 2009*; *Vance, 2010*; *Lu et al., 2015*). Because of the profound effect of this drug on all cells and thus anemone physiology, severe effects are to be expected. Accordingly, we found that both symbiotic and aposymbiotic anemones appear to lose tissue and shorten their tentacles in a dose- and duration-dependent manner. However, symbiotic anemones showed such effects on host physiology faster than their aposymbiotic counterparts (*Figure 3G*, *Figure 3—figure supplement 6*). Moreover, symbiont density decreased in response to U18666A treatment (*Figure 3H*). We observed similar effects with *A. digitifera* juvenile primary polyps stably hosting *Symbiodiniaceae* strain SSB01 when exposed to increasing concentrations of U18666A (*Figure 3—figure supplement 7*). This suggests that inhibition of sterol transport affects symbiosis stability and may lead to loss of symbionts ('bleaching'). Further, the disruption of global sterol transport compromises host tissues in all cases, emphasizing the importance of sterols in tissue homeostasis.

To test sterol-binding properties of *Aiptasia* NPC2, we compared the most conserved canonical NPC2 to the non-canonical NPC2 most up-regulated upon symbiosis (XM_021041171 to XM_021052404, respectively). We used lipidomics to quantify lipids bound by immunoprecipitated native or recombinant NPC2s (*Figure 4*, *Figure 4—figure supplement 1*) (after *Li et al., 2010*). Recombinant proteins were expressed in HEK 293T cells, after which cell lysates were mixed with *Symbiodiniaceae* SSB01 homogenates at either neutral conditions (pH 7) or acidic conditions reflecting the lysosome/symbiosome (pH 5). Under both conditions, canonical and non-canonical NPC2:mCherry fusion proteins bound symbiont-produced sterols significantly above the background levels of the control, mCherry alone (*Figure 4A*). The relative proportions of bound sterols generally exhibited equilibrium levels with the corresponding symbiont homogenate (*Figure 4B*). To validate sterol binding by non-canonical NPC2 in vivo, we also immunoprecipitated the native non-canonical NPC2 and bound sterols directly from homogenates of symbiotic *Aiptasia*. Again, we detected symbiont-produced sterols above background levels, validating our heterologous system and indicating that these proteins bind sterols in vivo during symbiosis (*Figure 4C*). These data indicate that, despite their evolutionary divergence, both types of *Aiptasia* NPC2s have the conserved function of binding sterols in lysosomal-like environments. Although we cannot rule out subtle differences in sterol binding dynamics between the two proteins, our results suggested no differential binding between canonical and non-canonical NPC2s, consistent with the observations that the sterol ligand and the residues lining the binding cavity tolerate considerable variations (*Xu et al., 2007*; *Liou et al., 2006*). Corroborating this, we were unable to detect any difference in the differential expression of canonical and non-canonical NPC2s between aposymbiotic and symbiotic state in three symbiont-*Aiptasia* pairings (*Figure 4D*).

With data suggesting both NPC2 types can bind symbiont-produced sterols, we were therefore left with the question: what is the functional advantage of localizing non-canonical NPC2s specifically in the symbiosome? The mature symbiosome, where non-canonical NPC2 appears to function, remains poorly understood; however, extreme acidity appears to be a unique characteristic of these specialized cellular compartments. Whereas lumenal pH of classic lysosomes can range from 4.7 to 6 (*Johnson et al., 2016*), recent work indicates that mature symbiosomes in steady-state symbiosis are even more acidic (pH ~4) to promote efficient photosynthesis (*Barott et al., 2015*). We therefore sought to compare the stability/solubility of representative canonical and non-canonical NPC2s at different pH (*Figure 4E + F*, *Figure 4—figure supplement 2*). Interestingly, the patterns of extracted soluble proteins vary among NPC2s: canonical NPC2 appears in one predominant form at both pH's (*Figure 4E*, red arrowhead), whereas one of the symbiosis-responsive non-canonical

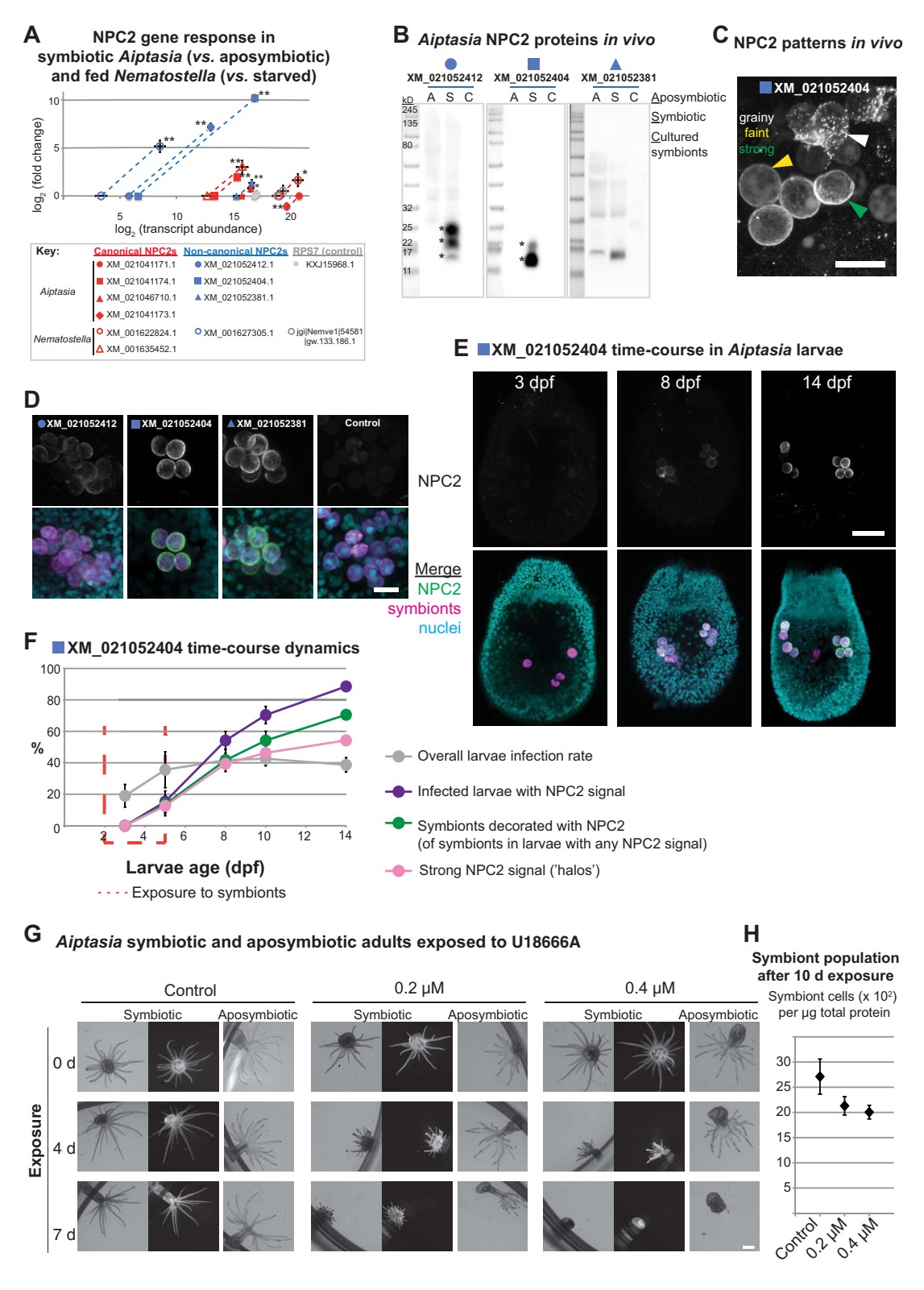

**Figure 3.** Non-canonical NPC2s respond to symbiosis and are spatiotemporally regulated to mature symbiosomes. (**A**) Gene expression by RT-qPCR of canonical (red symbols) and non-canonical (blue symbols) NPC2s and 40S ribosomal subunit (RPS7, gray symbols). Filled symbols: *Aiptasia*. Open symbols: *Nematostella*. Average values ± SD (error bars). Statistical comparisons by Bayesian modeling (see Materials and methods), *p<0.05, **p<0.005. (**B**) Homogenates of aposymbiotic (A) or symbiotic (S) *Aiptasia* adults and cultured *Symbiodiniaceae* strain SSB01 (C) detected with affinity-

*Figure 3 continued on next page*

*Figure 3 continued*

purified antibodies to non-canonical *Aiptasia* NPC2 homologues. Asterisks, NPC2 glycoforms. (**C**) Immunofluorescence (IF) patterns of non-canonical NPC2 in 14 d post-fertilization (dpf) *Aiptasia* larvae containing intracellular symbionts of *Symbiodiniaceae* strain SSB01. Scale bar, 10 μm. (**D**) IF of several non-canonical *Aiptasia* NPC2s as in **C**. Merge channels: NPC2, secondary antibody Alexa488-anti-rabbit IgG; Nuclei, Hoechst; Symbionts, red autofluorescence of photosynthetic machinery. Control, secondary antibody only. Scale bar, 10 μm. (**E**) Time-course of immunofluorescence of non-canonical NPC2 in *Aiptasia* larvae infected with *Symbiodiniaceae* SSB01 from 2-5 dpf. Larvae oral opening facing up. Merge as in **D**. Scale bar, 25 μm. (**F**) Quantification of NPC2 IF time-course in **E**. Average value ± SEM (error bars). (**G**) Brightfield and fluorescence micrographs of symbiotic and aposymbiotic *Aiptasia* exposed to U18666A or DMSO negative control (vol. equiv. to 10 μM addition). Symbiont red autofluorescence as above. Scale bar, 1 mm. (**H**) Quantification of symbiont density in symbiotic anemones from **G**. Average values ± SEM (error bars).

DOI: https://doi.org/10.7554/eLife.43923.008

The following figure supplements are available for figure 3:

**Figure supplement 1.** Validation of new antibodies raised against *Aiptasia* non-canonical NPC2s.
DOI: https://doi.org/10.7554/eLife.43923.009

**Figure supplement 2.** Differential maternal loading of canonical and non-canonical NPC2 transcripts in embryos of *Acropora* and *Nematostella*.
DOI: https://doi.org/10.7554/eLife.43923.010

**Figure supplement 3.** Intracellular symbiont surrounded by non-canonical NPC2.
DOI: https://doi.org/10.7554/eLife.43923.011

**Figure supplement 4.** Dynamic recruitment of other non-canonical NPC2s to intracellular symbionts increases as symbiosis matures.
DOI: https://doi.org/10.7554/eLife.43923.012

**Figure supplement 5.** Quantification of symbiont load in *Aiptasia* larvae in a time-course of non-canonical NPC2 IF.
DOI: https://doi.org/10.7554/eLife.43923.013

**Figure supplement 6.** All symbiotic and aposymbiotic animals in the U18666A exposure experiment in *Figure 3G + H*.
DOI: https://doi.org/10.7554/eLife.43923.014

**Figure supplement 7.** *Acropora digitifera* juvenile primary polyps hosting *Symbiodiniaceae* strain SSB01 exposed to U18666A or negative control.
DOI: https://doi.org/10.7554/eLife.43923.015

NPC2s (XM_021052412; *Figure 3A + B*) always appears in two forms in both conditions (*Figure 4E*, arrowheads and asterisks). Strikingly, the pattern for the other symbiosis-specific non-canonical NPC2 (XM_021052404; *Figures 3A–G* and *4A–C*) is distinct between pH 7 and pH 5, with a consistently occurring additional band at higher pH (*Figure 4E*, blue arrow). Although we cannot rule out that the additional bands reflect degradation products, we favor the interpretation that they most likely represent distinct glycoforms, which also occur in vivo (*Figure 3B*). When quantifying the protein variant common to all samples (*Figure 4E*, arrowheads), we found that at pH 5, the non-canonical NPC2s were consistently more abundant in the soluble fraction than the canonical counterpart (*Figure 4F*). For all proteins tested, the ratio of the predominant soluble protein variant at pH 7 to that at pH 5 was always >1, indicating more solubility at pH 7. However, the ratio was higher for *Aiptasia* canonical NPC2 than for the non-canonical NPC2s, indicating that the former is relatively less soluble at pH 5 (*Figure 4F*). Taken together, symbiosis-responsive non-canonical NPC2 appears to be more soluble/stable than canonical NPC2 at a lower pH, likely characteristic of the symbiosome. In line with this, all *Aiptasia* non-canonical NPC2 proteins harbored glycosylation sites and a glycine followed by a histidine residue (*Figure 2B*), which may contribute to protein stability in acidic environments (*Rudd et al., 1994*; *Hanson et al., 2009*; *Culyba et al., 2011*). However, pH-dependent protein stability is difficult to predict and functional experiments are required to determine whether such motifs (or others) play a role for the adaptation to the symbiosome or not.

In summary, our data reveal that the transfer of complex mixtures of symbiont-derived sterols is a key feature of anthozoan photosymbiosis (*Figure 4G*), whereby the specific composition and proportion of transferred sterols appears to be under the control of both symbiont and host. While the non-canonical NPC2 sterol-binding proteins are part of the machinery transferring sterols from symbiont to host, they do not contribute to host sterol selection by differential expression or differential binding. Instead, our assays reveal the possibility of an increased tolerance to acidic conditions of non-canonical NPC2s and their late accumulation in the symbiosome, consistent with gradual enrichment upon increasing symbiosome acidification. We propose that whereas ubiquitously expressed canonical NPC2 homologues are 'workhorses' in sterol trafficking throughout the host, non-canonical NPC2s are spatiotemporally regulated to accumulate as the symbiosome matures, developing into a unique compartment optimized to promote the interaction and communication of the symbiotic

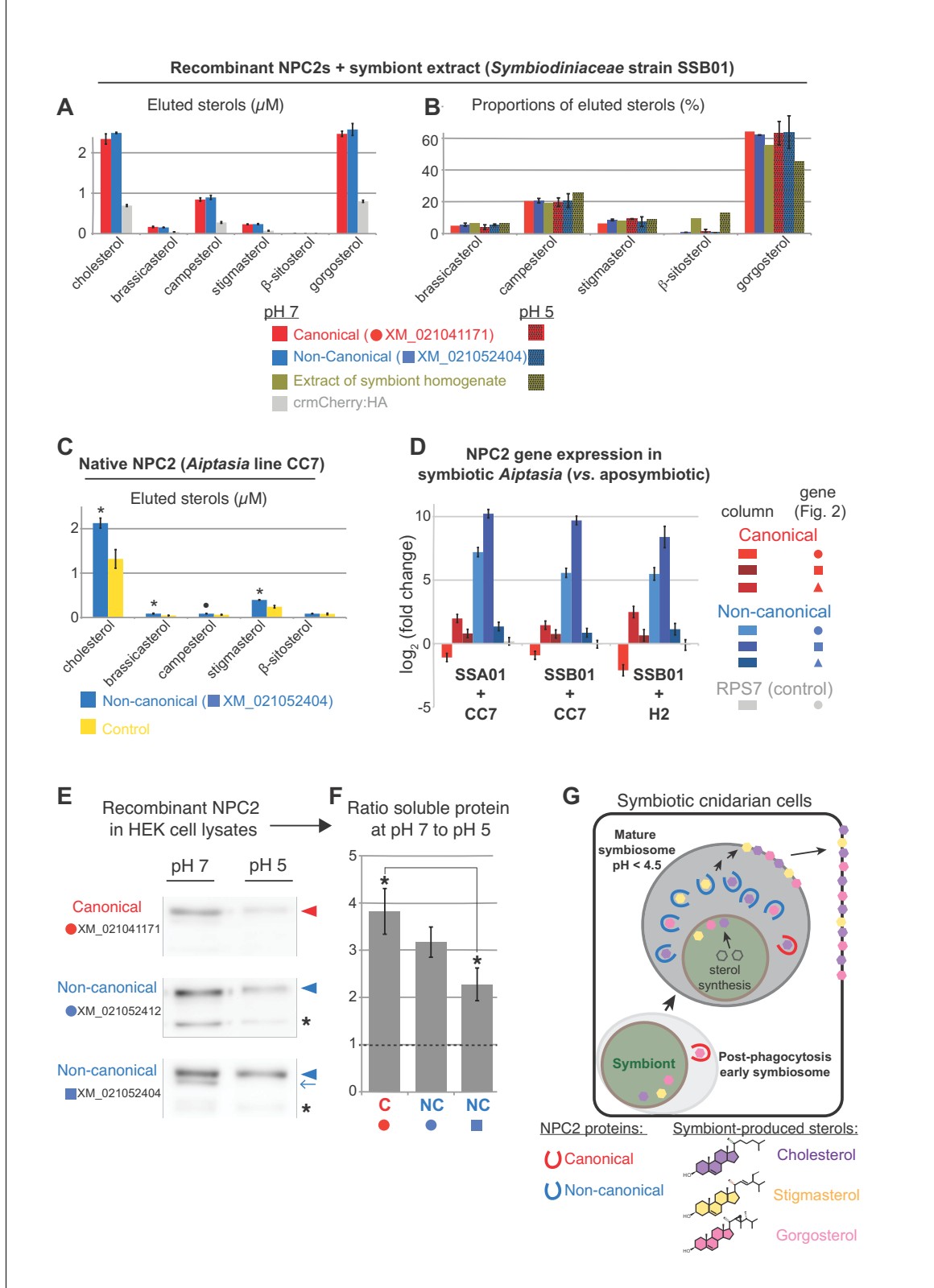

**Figure 4.** NPC2 binding to symbiont-produced sterols via immunoprecipitation-lipidomics, and differential stability of non-canonical NPC2 at varying pH. (**A**) Quantification of bound lipids in the eluates following IP of recombinant canonical and non-canonical NPC2:crmCherry:HA and negative control crmCherry:HA at pH 7 (*Figure 4—figure supplement 1*). Sterols comprising <1.5% were omitted for clarity. crmCherry, lysosome-stable cleavage-resistant mCherry (*Huang et al., 2014*) . Average values ± SD (error bars). Except ß-sitosterol, each NPC2 was significantly different to crmCherry

*Figure 4 continued on next page*

*Figure 4 continued*

negative control (Student's *t*-test, p<0.01). (**B**) Relative proportions of NPC2-bound sterols and the corresponding symbiont extract at pH 5 and 7. Average values ± SD (error bars) (**C**) Immunoprecipitation (IP) of native non-canonical NPC2 from *Aiptasia* and quantification of eluted bound sterols. Control, identical reaction omitting antibody. Average values ± SEM (error bars). Statistical comparisons to control (Student's *t*-test: *p<0.05, •p<0.09). (**D**) NPC2 gene expression by qPCR in the various *Aiptasia*/*Symbiodiniaceae* host/symbiont combinations in *Figure 1B*. (**E**) Recombinant NPC2s detected by mCherry antibody in the soluble fractions of HEK cell lysates. Lysate preparations were identical except for buffer pH; equivalent volumes loaded per lane. (**F**) Quantification of protein abundances from Western blots in **E**. Average values ± SEM (error bars). Student's *t*-test, *p<0.05. (**G**) We propose a model in which symbiotic anthozoans have evolved non-canonical NPC2 homologues that are spatiotemporally regulated to specifically respond to symbiosis, including through adaptation to the acidic environment of the symbiosome, the lysosomal-like organelle in which symbionts reside. NPC2 proteins bind and transport symbiont-produced sterols, and such trafficking is essential for cellular homeostasis of the sterol-auxotrophic hosts.

DOI: https://doi.org/10.7554/eLife.43923.016

The following figure supplements are available for figure 4:

**Figure supplement 1.** Immunoprecipitation-lipidomics: protein inputs and immunoprecipitation (IP) elutions.
DOI: https://doi.org/10.7554/eLife.43923.017

**Figure supplement 2.** Soluble recombinant NPC2s in HEK 293T cell lysates.
DOI: https://doi.org/10.7554/eLife.43923.018

partners (*Figure 4G*). This allows symbiotic cnidarians to flexibly use symbiont-produced sterols, with reef-building corals nearly fully substituting these for prey-derived cholesterol, supporting survival in nutrient-poor environments. More broadly, our findings indicate that carbon acquisition by lipid transfer, similar to other symbioses (*Keymer et al., 2017*), is a major driver of coral-algal symbiotic relationships as a means to adapt to various ecological niches by efficient exploitation of limited resources.

# Materials and methods

## Key resources table

| Reagent type (species) or resource | Designation | Source or reference | Identifiers | Additional information |
|---|---|---|---|---|
| Gene (*Exaiptasia pallida*) | *Aiptasia* canonical NPC2 XM_021041171.1 | NA | XM_021041171.1 | |
| Gene (*E. pallida*) | *Aiptasia* canonical NPC2 XM_021041174.1 | NA | XM_021041174.1 | |
| Gene (*E. pallida*) | *Aiptasia* canonical NPC2 XM_021041173.1 | NA | XM_021041173.1 | |
| Gene (*E. pallida*) | *Aiptasia* canonical NPC2 XM_021046710.1 | NA | XM_021046710.1 | |
| Gene (*E. pallida*) | *Aiptasia* non-canonical NPC2 XM_021052412.1 | NA | XM_021052412.1 | |
| Gene (*E. pallida*) | *Aiptasia* non-canonical NPC2 XM_021052404.1 | NA | XM_021052404.1 | |
| Gene (*E. pallida*) | *Aiptasia* non-canonical NPC2 XM_021052381.1 | NA | XM_021052381.1 | |
| Gene (*E. pallida*) | *Aiptasia* RPS7 | NA | KXJ15968.1 | |
| Gene (*Nematostella vectensis*) | *Nematostella* canonical NPC2 XM_001622824.1 | NA | XM_001622824.1 | |

*Continued on next page*

*Continued*

| Reagent type (species) or resource | Designation | Source or reference | Identifiers | Additional information |
|---|---|---|---|---|
| Gene (*N. vectensis*) | *Nematostella* canonical NPC2 XM_001635452.1 | NA | XM_001635452.1 | |
| Gene (*N. vectensis*) | *Nematostella* non-canonical NPC2 XM_001627305.1 | NA | XM_001627305.1 | |
| Gene (*N. vectensis*) | *Nematostella* RPS7 | NA | jgi\|Nemve1\|54581\|gw.133.186.1 | |
| Gene (*Acropora digitifera*) | *Acropora* canonical NPC2 XM_015898926.1 | NA | XM_015898926.1 | |
| Gene (*A. digitifera*) | *Acropora* canonical NPC2 XM_015898919.1 | NA | XM_015898919.1 | |
| Gene (*A. digitifera*) | *Acropora* canonical NPC2 XM_015910954.1 | NA | XM_015910954.1 | |
| Gene (*A. digitifera*) | *Acropora* canonical NPC2 XM_015898921.1 | NA | XM_015898921.1 | |
| Gene (*A. digitifera*) | *Acropora* non-canonical NPC2 XR_001561549.1 | NA | XR_001561549.1 | |
| Gene (*A. digitifera*) | *Acropora* non-canonical NPC2 AD_NPC2_6 | NA | AD_NPC2_6 | see *Supplementary file 1* |
| Gene (*A. digitifera*) | *Acropora* non-canonical NPC2 AD_NPC2_7 | NA | AD_NPC2_7 | see *Supplementary file 1* |
| Gene (*A. digitifera*) | *Acropora* RPS7 | NA | XM_015898841.1 | |
| Gene (*Cladocopium goreaui*) | *Acropora digitifera* resident *Symbiodiniaceae* symbionts, cp23S marker | NA | MK696595 | |
| Gene (*C. goreaui*) | CCMP2466 *Symbiodiniaceae* culture, cp23S marker | NA | MK696599 | |
| Strain, strain background (*Exaiptasia pallida*)(male) | *Aiptasia* line CC7 | DOI: 10.1038/srep15677 | | in DOI: 10.1186/1471-2164-10-258 |
| Strain, strain background (*E. pallida*)(female) | *Aiptasia* line F003 | DOI: 10.1038/srep15677 | | |
| Strain, strain background (*E. pallida*)(female) | *Aiptasia* line H2 | DOI: 10.1038/srep15677 | | |
| Strain, strain background (*Nematostella vectensis*) (male and female) | *Nematostella* | Prof. Dr. Thomas Holstein, Heidelberg University | | |
| Strain, strain background (*Breviolum minutum*) | *Symbiodiniaceae* strain SSB01 | DOI: 10.1111/jpy.12055 | GenBank: MK692539 | Accession number for rDNA 28S LSU marker (DOI: 10.1016/j.cub.2018.07.008) |

*Continued on next page*

*Continued*

| Reagent type (species) or resource | Designation | Source or reference | Identifiers | Additional information |
|---|---|---|---|---|
| Strain, strain background (*Symbiodinium linuchae*) | *Symbiodiniaceae* strain SSA01 | DOI: 10.1038/srep15677 | GenBank: MK692538 | Accession number for rDNA 28S LSU marker (DOI: 10.1016/j.cub.2018.07.008) |
| Strain, strain background (*Symbiodinium necroappetens*) | *Symbiodiniaceae* strain SSA02 | DOI: 10.1111/jpy.12055 | GenBank: MK692866 | Accession number for rDNA 28S LSU marker (DOI: 10.1016/j.cub.2018.07.008) |
| Strain, strain background (*Effrenium voratum*) | *Symbiodiniaceae* strain SSE01 | DOI: 10.1111/jpy.12055 | GenBank: MK696597 | Accession number for rDNA 28S LSU marker (DOI: 10.1016/j.cub.2018.07.008) |
| Strain, strain background (*Cladocopium goreaui*) | *Symbiodiniaceae* strain CCMP2466 | National Center for Marine Algae and Microbiota (NCMA), Bigelow Laboratory for Ocean Sciences, Maine, USA | GenBank: MK696600 | Accession number for rDNA 28S LSU marker (DOI: 10.1016/j.cub.2018.07.008) |
| Strain, strain background (*Durusdinium trenchii*) | *Symbiodiniaceae* strain CCMP2556 | National Center for Marine Algae and Microbiota (NCMA), Bigelow Laboratory for Ocean Sciences, Maine, USA | GenBank: MK692915 | Accession number for rDNA 28S LSU marker (DOI: 10.1016/j.cub.2018.07.008) |
| Cell line (*Homo sapiens*) | HEK 293T | Deutsche Sammlung von Mikroorganismen und Zellkulturen (DMSZ) from Dr. C. Niehrs and Dr. S. Acebrón | ACC 305 | authenticated by DMSZ, confirmed mycoplasma-free |
| Biological sample (*Acropora digitifera*) | *Acropora digitifera* | this paper | | Collected off Sesoko Island, Okinawa Prefecture, Japan (26°37′41"N, 127°51′38"E) |
| Antibody | anti-Aiptasia NPC2 XM_021052412 | this paper | | Rabbit polyclonal, 0.3 mg/ml. (Westerns 1:500-1:1000, Dot Blot 1:1000, IF 1:200) BioScience GmbH. |
| Antibody | anti-Aiptasia NPC2 XM_021052404 | this paper | | Rabbit polyclonal, 0.45 mg/ml. (Westerns 1:2000-1:4000, Dot Blot 1:5000, IF 1:100-1:750) BioScience GmbH. |
| Antibody | anti-Aiptasia NPC2 XM_021052381 | this paper | | Rabbit polyclonal, 0.4 mg/ml (Westerns 1:500, Dot Blot 1:500, IF 1:200) BioScience GmbH. |
| Antibody | HRP-coupled anti-rabbit (goat polyclonal) | Jackson ImmunoResearch | Jackson ImmunoResearch: 111-035-144 | (Western blot 1:10,000) |
| Antibody | anti-alpha-tubulin (mouse monoclonal) | Sigma-Aldrich | Sigma-Aldrich:T9026 | (Western blot 1:1000) |

*Continued on next page*

*Continued*

| Reagent type (species) or resource | Designation | Source or reference | Identifiers | Additional information |
|---|---|---|---|---|
| Antibody | HRP-coupled anti-mouse (goat polyclonal) | Jackson ImmunoResearch | Jackson ImmunoResearch: 115-035-044 | (Western blot 1:10,000) |
| Antibody | anti-rabbit IgG-Alexa488 (goat polyclonal) | Abcam | Abcam:ab150089 | (IF 1:500) |
| Antibody | anti-mCherry (rabbit polyclonal) | Thermo Fisher Scientific | Thermo Fisher Scientific:PA5-34974 | (Western blot 1:3000) |
| Antibody | conformation-specific HRP-coupled anti-rabbit IgG (mouse monoclonal) | Cell Signaling Technology | CST:5127S | (Western blot 1:2000) |
| Recombinant DNA reagent | NPC2 XM_021052412 for recombinant expression in HEK293T cells (plasmid) | this paper | | NPC2-5P-crmCherry (DOI: 10.1371/journal.pone.0088893)−3xHA (YPYDVPDYA). Progenitors: PCR (cDNA), vector pCEP |
| Recombinant DNA reagent | NPC2 XM_021052404 for recombinant expression in HEK293T cells (plasmid) | this paper | | NPC2-5P-crmCherry (DOI: 10.1371/journal.pone.0088893)−3xHA (YPYDVPDYA). Progenitors: PCR (cDNA), vector pCEP |
| Recombinant DNA reagent | NPC2 XM_021041171 for recombinant expression in HEK293T cells (plasmid) | this paper | | NPC2-5P-crmCherry (DOI: 10.1371/journal.pone.0088893)−3xHA (YPYDVPDYA). Progenitors: PCR (cDNA), vector pCEP |
| Recombinant DNA reagent | crmCherry:3xHA control for recombinant expression in HEK293T cells (plasmid) | this paper | | crmCherry (DOI: 10.1371/journal.pone.0088893)−3xHA (YPYDVPDYA). Progenitors: PCR (cDNA), vector pCEP |
| Sequence-based reagent | Primers for qPCR of *Aiptasia, Acropora, Nematostella* NPC2s | this paper | | see *Supplementary file 3* for all primer sequences |
| Peptide, recombinant protein | K-YGIDVFCDEIRIHLT | Custom peptide, INTAVIS Bioanalytical Instruments AG | | Epitope for antibody against Aiptasia NPC2 XM_021052412 |
| Peptide, recombinant protein | K-AKNDIFCNSIPFNLV | Custom peptide, INTAVIS Bioanalytical Instruments AG | | Epitope for antibody against Aiptasia NPC2 XM_021052404 |
| Peptide, recombinant protein | K-VQNNVLCGEVTLTLM | Custom peptide, INTAVIS Bioanalytical Instruments AG | | Epitope for antibody against Aiptasia NPC2 XM_021052381 |
| Commercial assay or kit | RNeasy kit | Qiagen | Qiagen:74104 | |
| Commercial assay or kit | SYBR Hi-ROX qPCR master mix | Bioline | BIO-92005 | |
| Commercial assay or kit | NHS-activated Sepharose Fast Flow 4 | GE Health Care Life Sciences | GE Healthcare Sciences:17090601 | |

*Continued on next page*

*Continued*

| Reagent type (species) or resource | Designation | Source or reference | Identifiers | Additional information |
|---|---|---|---|---|
| Commercial assay or kit | anti-HA magnetic beads | Miltenyi Biotech | Miltenyi Biotech: 130-091-122 | |
| Commercial assay or kit | Dynabeads Antibody Coupling Kit | Thermo Fisher Scientific | Thermo Fisher Scientific: 14311 | |
| Commercial assay or kit | Pierce BCA Protein Assay Kit | Thermo Fisher Scientific | Thermo Fisher Scientific: 23227 | |
| Chemical compound, drug | Trizol | Life Technologies | Life Technologies: 15596026 | |
| Chemical compound, drug | MSTFA (*N*-Methyl-*N*-(trimethylsilyl) trifluoroacetamide | Sigma-Aldrich | Sigma-Aldrich:69479 | |
| Chemical compound, drug | Lipofectamine2000 | Thermo Fisher Scientific | Thermo Fisher Scientific:11668019 | |
| Chemical compound, drug | Cholesterol-D6 | Cambridge Isotope Laboratories | Cambridge Isotope Laboratories: DLM2607 | |
| Chemical compound, drug | Acetyl chloride in methylene chloride | Sigma-Aldrich | Sigma-Aldrich:708496 | |
| Chemical compound, drug | U18666A | Sigma-Aldrich | Sigma-Aldrich:U3633 | |
| Chemical compound, drug | RNAlater | Thermo Fisher Scientific | Thermo Fisher Scientific:AM7020 | |
| Software, algorithm | Geneious | DOI: 10.1093/bioinformatics/bts199 | | v. 9 |
| Software, algorithm | SignalP 4.0 | DOI: 10.1038/nmeth.1701 | | v. 4.0 |
| Software, algorithm | MEGA | DOI: 10.1093/molbev/msw054 | | v. 7.10.8 |
| Software, algorithm | MrBayes | DOI: 10.1093/bioinformatics/17.8.754 | | v. 3.2.6; plugin for Geneious |
| Software, algorithm | DataMonkey server | DOI: 10.1093/bioinformatics/bti320 | | Datamonkey classic server |
| Software, algorithm | HyPhy program suite | DOI: 10.1093/bioinformatics/bti079 | | accessed via DataMonkey classic server |
| Software, algorithm | single-likelihood ancestor counting (SLAC) | DOI: 10.1093/molbev/msi105 | | accessed via DataMonkey classic server |
| Software, algorithm | mixed effects models of evolution (MEME) | DOI: 10.1371/journal.pgen.1002764 | | accessed via DataMonkey classic server |
| Software, algorithm | MCMC.qPCR | DOI: 10.1371/journal.pone.0071448 | | R library |
| Software, algorithm | GCMS Postrun Analysis software | Shimadzu | | |
| Software, algorithm | Analyst | SCIEX | | v. 1.6.3. Control and analysis software for QTRAP 5500 MS |
| Software, algorithm | LipidView | SCIEX | | v. 1.2 |
| Software, algorithm | Fiji | DOI: 10.1038/nmeth.2019 | | v. 2.0.0-rc-67/1.52d |

*Continued on next page*

*Continued*

| Reagent type (species) or resource | Designation | Source or reference | Identifiers | Additional information |
|---|---|---|---|---|
| Other | Phalloidin; Phalloidin-Atto 565 | Sigma-Aldrich | Sigma-Aldrich:94072 | |
| Other | Hoechst; Hoechst 33258 | Sigma-Aldrich | Sigma-Aldrich:B2883 | |

## Computational methods

### NPC2 bayesian consensus phylogeny construction

Genomes and, if available, proteomes and transcriptomes (*Supplementary file 2*) were loaded into Geneious v.9 (*Kearse et al., 2012*). Proteomes and transcriptomes were searched with BLASTp and BLASTx (both v.2.8.0), respectively, with NPC2 homologues from *Aiptasia*, human, and related taxa as queries. Genomic loci were identified via discontinous Megablast. The top NPC2 BLASTp hits in the single-celled eukaryotic filasterian *Capsaspora owczarzaki* included two homologues of phospho-lipid transfer protein. With similar sizes to NPC2 and a shared predicted ML superfamily domain, these were included in analyses and one (XP_004344261.1) used as an outgroup during phylogenetic tree construction. Signal peptides were predicted using the SignalP4.0 server (*Petersen et al., 2011*) and, together with stop codons, removed from further analyses. 77 NPC2 homologue sequences were aligned by codon using MUSCLE with default parameters and manually trimmed in MEGA7 (v7.10.8) (*Kumar et al., 2016*), where the best model was calculated as GTR+G. Bayesian phylogenies were inferred using MrBayes v.3.2.6 (*Huelsenbeck and Ronquist, 2001*) plugin in Geneious, with the GTR model, gamma rate variation, and five gamma categories. The consensus tree was estimated from four chains (temperature 0.2) for 1,000,000 generations, sampling every 200th tree after 25% burn-in.

### Adaptive evolution

Evidence of selection was calculated using the DataMonkey server (http://classic.datamonkey.org) for the HyPhy program suite (*Pond et al., 2005*; *Pond and Frost, 2005*). Briefly, *Aiptasia* and *Nematostella* canonical and non-canonical NPC2 sequences were aligned by codon using MUSCLE in MEGA7 as above, and the best substitution models calculated. Bayesian phylogenies were inferred with MrBayes as above except for the following parameters: GTR+G+I, four gamma categories, 50,000 generations and sampling every 100th tree after 20% burn-in. Trees were uploaded on the DataMonkey server and analysed with: i) fixed effects likelihood (FEL); ii) random effects likelihood (REL); iii) single-likelihood ancestor counting (SLAC) (*Kosakovsky Pond and Frost, 2005*); and iv) mixed effects model of evolution (MEME) (*Murrell et al., 2012*), and results were concatenated with the 'Integrative Selection Analysis' tool.

## Live organism culture and collection

### *Aiptasia* adults

 *Aiptasia* were cultured as described (*Grawunder et al., 2015*); animals rendered aposymbiotic (*Matthews et al., 2016*) were kept so for over one year before experimentation. Animals were fed three times weekly with *Artemia* brine shrimp nauplii, shown to contain only cholesterol (*Tolosa et al., 2011*), and were starved for at least four weeks prior to sampling. For sampling, animals were removed from their tanks simultaneously around mid-day, blotted briefly on lab tissue to remove excess seawater, and then prepared for either qPCR or GC/MS. For qPCR, animals were added to 1 ml Trizol (15596026, Life Technologies), after which they were quickly homogenized with a homogenizer (Miccra D-1, Miccra GmbH) at setting 3 for 10–15 s and then frozen at −80°C until RNA extraction. For GC/MS, animals were added to 400 µl ultrapure water, homogenized, and immediately processed. For GC/MS separation experiments, animal homogenates were centrifuged at 800x*g* for 5 min, after which the supernatant was separated and the pellet resuspended in 400 µl ultrapure water and immediately processed. Cells in the supernatant and pellet fractions were quantified with a visual particle counter (TC20, BioRad).

## *Aiptasia* eggs and larvae

Adults of strains F003 and CC7 were induced to spawn as described (*Grawunder et al., 2015*). For GC/MS, approx. 1000–3000 unfertilized eggs from female-only tanks were collected gently with transfer pipette within 2 hr of spawning, washed quickly in water and then in methanol, and resuspended in 750 µl methanol. For NPC2 immunofluorescence (IF) during symbiosis establishment, *Aiptasia* larvae 2 days post-fertilization (dpf) at a density of 300–500/ml FASW were exposed to *Symbiodiniaceae* strain SSB01 as described (*Bucher et al., 2016*) at 10,000 cells/ml. Larvae and algae were co-cultivated for 3 d, until at five dpf the larvae were filtered, washed, and resuspended in fresh FASW at a density of 300–500/ml. Larvae were fixed at the indicated time-points with 4% formaldehyde in filtered artificial seawater (FASW) rotating for 45 min at RT, washed twice with PBT (1x PBS pH 7.4 + 0.2% Triton-X), and stored in PBS at 4°C in the dark.

## *Nematostella* adults

For qPCR, mixed-sex *Nematostella* were kept in 12:12 L:D at 26°C and fed weekly with *Artemia* nauplii. Animals were then separated and either starved or fed *Artemia* nauplii daily for 14 d with subsequent daily water changes; animals were then starved for a further 2 d and then sampled as for *Aiptasia* qPCR. For GC/MS, mixed-sex *Nematostella* were kept in constant dark at 16°C, fed once weekly with *Artemia* nauplii, and water changed the following day; animals were starved for 10 d and then sampled as for *Aiptasia* GC/MS.

## *Acropora digitifera* adults, larvae, and primary polyps

Colonies of the coral *Acropora digitifera* were collected off Sesoko Island (26°37'41"N, 127°51'38"E, Okinawa, Japan) according to Okinawa Prefecture permits and CITES export and import permits (T-WA-17–000765). Corals were kept as described (*Wolfowicz et al., 2016*) at Sesoko Tropical Biosphere Research Center (University of Ryukyus, Okinawa, Japan). Colonies were isolated prior to spawning, and subsequently-spawned bundles of symbiont-free gametes were mixed for fertilization of defined crosses. The resulting planula larvae were maintained at approximately 1000 larvae/L in 10µm-filtered natural seawater (FNSW) exchanged daily. For GC/MS, samples were collected from adult parental colonies and their embryo offspring 19 and 24 hr post-fertilization (hpf), respectively, and immediately transferred to methanol. For qPCR, adults and their embryo offspring were simultaneously collected at the indicated hpf and immediately transferred into RNAlater (AM7020, Thermo Fisher Scientific). Samples were transferred to 4°C within hours and to −20°C within 2 d, where they were kept until processing. To generate juvenile primary polyps, larvae were induced to settle at six dpf and infected with *Symbiodiniaceae* strain SSB01 as described (*Wolfowicz et al., 2016*) for 4 d. Resident *Symbiodiniaceae* in adult parental colonies were typed with the chloroplast ribosomal DNA subunit 23S marker (cp23S) as previously described (*Grawunder et al., 2015*): 10 bacterial clones were sequenced per coral colony and all were identical (GenBank Accession MK696595), identified by BLASTn to the nr NCBI database as *Symbiodiniaceae* Clade C1.

## *Symbiodiniaceae* cultures

Clonal and axenic *Symbiodiniaceae* strains were typed with the 28S large ribosomal subunit marker as described (*LaJeunesse et al., 2018*) (organism, GenBank Accession): SSB01 (*Breviolum minutum*, MK692539), SSA01 (*Symbiodinium linuchae*, MK692538), SSA02 (*Symbiodinium necroappetens*, MK692866), and SSE01 (*Effrenium voratum*, MK696597) (*Xiang et al., 2013*) as well as the non-clonal, non-axenic strains CCMP2466 (*Cladocopium goreaui*, MK696600) and CCMP2556 (*Durusdinium trenchii*, MK692915) purchased from the National Center for Marine Algae and Microbiota (NCMA, Bigelow Laboratory for Ocean Sciences, Maine, USA). All strains were cultured as described (*Tolleter et al., 2013*). For GC/MS, $2.6 \times 10^7$ cells were collected at mid-day by gentle centrifugation at RT, washed twice in FASW, and the cell pellet resuspended in ultrapure water and processed as described. For GC/MS separation experiments, $1.1 \times 10^7$ (SSA01) or $1.6 \times 10^7$ (SSB01) cells cultured in identical conditions as the anemones were used. Strain CCMP2466 was additionally typed with the cp23S marker as previously described (*Grawunder et al., 2015*); 10 bacterial clones contained an identical sequence (GenBank Accession 696599) with 1 bp different to that of the *Acropora digitifera* endogenous symbionts described above (GenBank Accession 696595).

## Cell culture

Cells were obtained from Dr. Christoph Niehrs and Dr. Sergio Acebrón from the Deutsche Sammlung von Mikroorganismen und Zellkulturen GmbH (DSMZ) as HEK-293 cell line ACC 305, authenticated by multiplex PCR and IEF, and confirmed mycoplasma-free. Cells were transformed with SV40 T-antigen to generate HEK 293T cells and again confirmed to be mycoplasma-free in 2017. Cells were cultured in 1X DMEM medium (41966029, Gibco/Thermo Scientific) with 10% FBS and 1% pen/strep (100 µg/ml final concentrations). Cells were grown at 37°C with 5% carbon dioxide and passaged regularly.

## Gene expression

### RNA extraction and qPCR

RNA was extracted according to a hybrid protocol (*Polato et al., 2011*) with phenol-chloroform and the RNeasy kit (74104, Qiagen). RNA was qualitatively and quantitatively assessed via gel electrophoresis and NanoDrop spectrophotometry (Nanodrop1000), respectively, aliquoted and flash frozen in liquid nitrogen and stored at −80C. First strand cDNA synthesis was performed with the ReadyScript cDNA synthesis kit (RDRT, Sigma Aldrich) according to the manufacturer's instructions. qPCR was performed in 96 well plate format, with each reaction containing 0.4 µm each primer, 50 ng cDNA, and 1X SensiFast SYBR Hi-ROX qPCR master mix (BIO-92005, Bioline) in 20 µl total; reactions were measured on a StepOnePlus (Applied Biosystems). The gene encoding 40S Ribosomal Protein S7 (RPS7) was chosen as a comparison/baseline gene due to its demonstrated stability in a previous study (*Lehnert et al., 2014*). Primers (*Supplementary file 3*) were validated by amplicon sequencing through either TOPO-TA cloning (450071, Thermo Fisher Scientific) according to the standard protocol or, for *Acropora* and *Nematostella*, direct sequencing of qPCR products, with at least three sequences per product. Melt curves performed after each qPCR run confirmed the existence of single products per reaction. Amplification efficiencies of each primer pair were determined by a 3- or 4-point dilution series. Output was analyzed with the Bayesian analysis pipeline MCMC.qPCR (*Matz et al., 2013*) run according to standard protocol (https://matzlab.weebly.com/data–code.html). For *Acropora* adults and embryos, the model was run with 'naïve' parameters. For comparative expression within symbiotic *Aiptasia* and fed *Nematostella*, the analysis was run with 'informed' parameters setting RPS7 as a reference gene. $Log_2$ (fold change) and $Log_2$ (transcript abundance) were determined from command 'HPDsummary' with and without 'relative = TRUE', respectively; p-values of differential expression were calculated with command 'geneWise' on the former.

### *Nematostella* embryonic development

Expression data on *Nematostella* embryonic development and comparative adults were obtained from NvERTx (*Warner et al., 2018*) (http://ircan.unice.fr/ER/ER_plotter/home). Transcripts were identified by BLAST search to the NvERTx database as the *NPC2* homologues XM_001622824.1 (NvERTx.4.51280); XM_001627305.1 (NvERTx.4.192779); XM_001635452.1 (NvERTx.4.142169), and the *RPS7* homologue jgi|Nemve1|54581|gw.133.186.1 (NvERTx.4.145315). Transcript abundance counts at 0 hpf (unfertilized) comprised duplicate samples of 300 embryos each (*Fischer et al., 2014*). As a baseline for typical gene expression in adults, transcript abundance counts of 'uncut controls' (UC) comprised triplicate samples of 300 untreated 6-week-old adults (*Warner et al., 2018*).

## Sterol profiling with gas chromatography/mass spectrometry (GC/MS)

Samples were extracted with a modified Bligh-Dyer method: briefly, either 300 µl aqueous *Aiptasia* or *Nematostella* homogenate was added to 750 µl HPLC-grade methanol, or 300 µl ultra-pure water was added to the *Acropora* sample already in 750 µl methanol or ethanol. After shaking at 70°C for 45 min, the mixture was extracted with 375 µl HPLC-grade chloroform and 300 µl ultra-pure water and centrifugation. The dried organic phase was then saponified with 500 µl of 5% KOH in a 9:1 methanol:water solution and incubating at 68°C for 1 hr. The mixture was then extracted with water and chloroform followed by centrifugation. Lipids in the dried organic phase were derivatized to trimethylsilyl ethers with 25–40 µl MSTFA (#69479, Sigma Aldrich) at 60°C for 0.5–1 hr and immediately analysed. 1 µl of each mixture was injected into a QP2010-Plus GC/MS (Shimadzu) and with a

protocol (adapted from *Schouten et al., 1998*) as follows: oven temperature 60°C, increase to 130°C at 20 °C/min, then increase to 300°C at 4 °C/min and hold for 10 min. Spectra were collected between m/z 40 and 850 and were analysed in GCMS PostRun Analysis Software (Shimadzu) by comparison to the National Institute of Standards and Technology 2011 database. Relative sterol composition as percent of total sterols were calculated from integrated peak intensity on the total ion chromatograph for each sample.

### *Aiptasia*-specific anti-NPC2 antibodies and testing by dot blot

Antibodies were raised against the peptides K-YGIDVFCDEIRIHLT (XM_021052412), K-AKNDIFCNSI PFNLV (XM_021052404), and K-VQNNVLCGEVTLTLM (XM_021052381) coupled to the adjuvant keyhole limpet hemocyanin in rabbits (BioScience GmbH). Antibodies were affinity-purified from the antisera using the synthetic peptides (INTAVIS Bioanalytical Instruments AG) coupled to NHS-Activated Sepharose Fast Flow 4 (17090601, GE Health Care Life Sciences) according to the manufacturer's protocols. In dot blots, peptides dissolved in DMSO or water were spotted onto nitrocellulose membranes and allowed to dry 1 hr in a dessicant chamber. Blots were blocked in 5% milk PBS-T for 2.5 hr at RT and then incubated at 4°C overnight with non-canonical NPC2 antibodies diluted in 5% milk PBS-T as follows: (XM_021052412 at 1:1000, XM_021052404 at 1:5000, and XM_021052381 at 1:500). Blots were then incubated with HRP-coupled anti-rabbit antibody and further processed as described below for 'Western blots'.

### Western blots of *Aiptasia* and *Symbiodiniaceae* homogenates

Two aposymbiotic or symbiotic adult *Aiptasia* polyps were homogenized in buffer A with 2X Halt Protease Inhibitor Cocktail (78430, Thermo Fisher Scientific) and then sonicated on ice (Sonifier 250, Branson Ultrasonics) with two rounds of 25 pulses at duty cycle 40%, output control 1.8. From cultured *Symbiodiniaceae* strain SSB01, $1.2 \times 10^7$ cells were collected by gentle centrifugation. After addition of buffer A and glass beads (425–600 µm), cells were disrupted by vortexing six times for 1 min each, with 1 min on ice in between each, then further disrupted by passage through a G23 needle. All homogenates were then centrifuged at 20,000x*g* for 10 min at 4°C, and three sets of identical volumes of the supernatants were resolved on a 12% Tricine-SDS-Page gel and transferred by Western blot onto nitrocellulose membranes. Membranes were blocked for 1 hr in 5% milk PBS-T and then incubated with antibodies raised against three different non-canonical *Aiptasia* NPC2s (XM_021052404 at 1:4000, XM_021052412 at 1:1000 and XM_021052381 at 1:500) in 5% milk PBS-T at 4°C overnight, followed by incubation with HRP-coupled anti-rabbit (Jackson ImmunoResearch) at 1:10000 in 5% milk PBS-T at RT for 1 hr, and then detection with ECL (GERPN2232, Sigma-Aldrich) and imaging on ECL Imager (ChemoCam, Intas). For peptide-blocked controls, 40 µg of homogenate supernatant per lane was resolved on a 10% Tris-tricine-SDS-Page gel and transferred and blocked as above. Antibodies were diluted in 5% milk PBS-T (XM_021052404 at 1:2000, XM_021052412 at 1:500 and XM_021052381 at 1:500) and the corresponding immunogenic peptides solubilized in DMSO or PBS at 0.5 mg/ml - 1 mg/ml were added at the indicated peptide:antibody (mass:mass) ratios. The peptide-antibody mixtures were rotated overnight at 4°C and then incubated with the blots at 4°C for approx. 60 hr, after which blots were incubated with anti-rabbit secondary antibody and processed as above. Blots were then re-blocked, incubated with anti-alpha-tubulin antibody (1:1000, T9026, Sigma-Aldrich), then HRP-coupled anti-mouse (1:10000, Jackson ImmunoResearch), and imaged as above.

### Immunoprecipitation-lipidomics of NPC2-sterol binding
#### Buffers
**A:** 200 mM Ammonium Acetate, pH 7; **B**: 200 mM Ammonium Acetate, pH 5; **C**: 50 mM MES, 150 mM NaCl, 0.004% Nonidet P-40; **D**: 50 mM Tris, 150 mM NaCl, 0.004% Nonidet P-40, pH 7.5.

#### Cell culture lysates and symbiont extracts

NPC2 proteins were cloned behind the cytomegalovirus promoter in a pCEP-based vector followed by a five-proline linker, cleavage-resistant mCherry (crmCherry) (*Huang et al., 2014*), and a 3xHA tag (YPYDVPDYA). A control vector contained only crmCherry:3xHA. Vectors were transiently transfected with Lipofectamine2000 (11668019, Invitrogen/Thermo Fisher Scientific) according to the

manufacturer's protocol into HEK 293T cells in 10 cm diameter dishes. After growth for 48 hr at 32°C, cells were rinsed with PBS and harvested in 1 ml of Buffer A, B, C, or D with Halt Protease Inhibitor Cocktail at 2X (78430, Thermo Fisher Scientific). Lysates were then sonicated on ice as above, centrifuged at 20,000xg for 20 min at 4°C, and supernatants used in binding assays. Approximately $2.5 \times 10^8$ cells of *Symbiodiniaceae* strain SSB01 approx. 7 d after passaging were collected by gentle centrifugation. Cells were washed in 10 ml of Buffer A, B, C, or D (per the corresponding HEK cell lysate), and then 5 ml buffer was added to the pellet and cells were sonicated twice for 5 min at duty cycle 80%, output control 3. During sonication, extracts were allowed to heat slightly but not boil. Extracts were centrifuged at 6000xg for 10 min at RT, and supernatants used in binding assays.

## Immunoprecipitation

Cell lysates were incubated with symbiont extracts (450 µl and 500 µl, respectively) for 30 min at room temperature rotating, after which 25 µl anti-HA beads (130-091-122, Miltenyi Biotech) were added and the mixtures incubated rotating at RT for a further 30 min. Mixtures were then passed through magnetic columns on a magnetic plate (130-042-701, Miltenyi Biotech) pre-rinsed with 200 µl of the corresponding Buffer A, B, C, or D. Bound material on the column was rinsed four times with Buffer A, B, C, or D with protease inhibitor, and then once with the corresponding buffer half-diluted and without protease inhibitor. Lipids were eluted by application of 20 µl HPLC-grade methanol to the column for 5 min incubation, followed by 100 µl methanol and collected into HPLC glass bottles with glass inserts and Rubber/PTFE caps (Neochrom, NeoLab). Eluates were immediately transferred to ice and then −20°C until lipidomics processing on the same day. Proteins were then eluted by application of 20 µl loading dye (20 mM DTT, 60 mM Tris pH 6.8, 20% glycerol, 1% SDS, 0.01% Bromophenol blue) at 100°C to the column for 5 min incubation, followed by 50 µl loading dye and collection. Samples were then heated to 95°C for 3 min and then immediately resolved by SDS PAGE on 10% Tris-Glycine gels and transferred by Western blot onto nitrocellulose membranes. Membranes were blocked for 1 hr in 5% milk TBS-T and then incubated with anti-mCherry (PA5-34974, Thermo Fisher Scientific) at 1:3000 in 5% milk TBS-T at 4°C overnight, followed by incubation with conformation-specific HRP-coupled anti-rabbit (5127S, Cell Signaling Technology) at 1:2000 in 5% milk TBS-T at RT for 2 hr, and then detection with ECL (GERPN2232, Sigma-Aldrich) and imaging on ECL Imager (ChemoCam, Intas).

## Immunoprecipitation from *Aiptasia* homogenates

Purified polyclonal antibody against XM_021052404 (described above) was coupled to epoxy magnetic beads in the Dynabeads Antibody Coupling Kit (14311, Thermo Fisher Scientific) per the manufacturer's instructions. Beads (1 mg per reaction) were incubated with *Aiptasia* CC7 homogenates for 16 hr rotating at 4°C; control reactions contained uncoupled beads. After washing, protein-lipid complexes were immunoprecipitated via magnetic separation and eluted from beads with 200 mM glycine, pH 2.3, then immediately neutralized with 0.1 M Tris-HCl, pH 8.5. An aliquot was taken for Western blot visualization of proteins; the remainder was extracted for lipids with a mixture of chloroform:methanol:water (final ratios 8:4:3). The dried organic phase was reconstituted with 100 µl methanol and collected into HPLC glass bottles with glass inserts and Rubber/PTFE caps (Neochrom, NeoLab). Eluates were immediately transferred to ice and then −20°C until lipidomics processing on the same day.

## Lipidomics

50 µl of each eluate was added to chloroform-rinsed glass tubes, followed by addition of 100 pmol cholesterol-D6 (DLM2607, Cambridge Isotope Laboratories) as an internal standard. Samples were dried under nitrogen, derivatized with addition of 50 µmol acetyl chloride in methylene chloride (708496, Sigma Aldrich) for 30 min at RT, and then dried under nitrogen again. Samples were finally dissolved in 100 µl of MS buffer (100 mM ammonium acetate in methanol). For analysis, samples were diluted 1 in 10 in MS buffer and loaded into 96-well plates (Z651400-25A, Sigma Aldrich) for analysis. A standard curve in duplicate of pmol cholesterol/stigmasterol at 50/25, 250/125, 500/250 was always processed in parallel. Samples were injected by a TriVersa NanoMate held at 10°C on positive polarity at 1.2 kV and 0.4 psi gas pressure through a D-Type nozzle chip with 4 µm nominal diameter. Samples were analysed on a QTRAP 5500 (SCIEX) Hybrid Triple-Quadrupole/Linear Ion

Trap Mass Spectrometry system (MS) including SelexIon Differential Ion Mobility System (DMS). Analysis was run at an interface heater temperature of 60°C in positive ion and neutral loss scan mode (loss of acetate, 77.05 Da), with low Q1 resolution and high Q3 resolution at a scan speed of 200 Da/sec and 120 multiple acquisitions in the mass range 400–600 Da. Samples were run with a declustering potential (DP) of 55 V, entrance potential of 10 V, collision energy of 13 V, and collision cell exit potential of 14 V. The DMS ran at 60°C, medium pressure (24 psi), and a compensation voltage (COV) of 4.4 kV for the set separation voltage (SV) of 4000 V. In every run a pooled mixture of all samples was run with a COV ramp from 0 to 20 kV to confirm the appropriate COV. The instrument was driven by Analyst software version 1.6.3 (SCIEX), and data evaluation was performed using the software LipidView 1.2 (SCIEX) to detect and quantify sterols by peak intensities. Sterol concentrations were calculated by normalization to the cholesterol-D6 internal standard, subtraction of blank samples, and comparison to the standard curve.

## Western blots of soluble NPC2 proteins at different pH

Supernatants of recombinant NPC2 proteins in 1 ml Buffer A or B were obtained as described above. Equal volumes of supernatant were mixed with loading dye and resolved by SDS PAGE and Western blotting as described for 'Immunoprecipitation'. Quantification was performed in Fiji (*Schindelin et al., 2012*): for each band, the integrated density (ID) in a rectangular region-of-interest (ROI) around the band was calculated, less the background (ID of the same ROI above the band).

## Immunofluorescence of NPC2 in *Aiptasia* larvae

Fixed larvae in PBS at 4°C were permeabilized in PBT for 2 hr at RT. Samples were then incubated in blocking buffer (5% normal goat serum and 1% BSA in PBT) overnight at 4°C and then with primary antibody diluted in block buffer for 4 hr at RT at the following concentrations: 4.5 µg/ml (XM_021052404), 1.5 µg/ml (XM_021052412), and 2 µg/ml (XM_021052381). Samples were then washed twice for 5 min with PBT at RT, twice for approx. 18 hr at 4°C, then incubated with secondary antibody (goat anti-rabbit IgG-Alexa488; ab150089, Abcam) diluted to 4 µg/ml in block buffer for approx. 5 hr at RT. Samples were then washed with PBT three times for 5 min each at RT, then approx. 18 hr at 4°C. When phalloidin staining was included, samples were then washed with 1% BSA in PBS and incubated with Phalloidin-Atto 565 (94072, Sigma-Aldrich) in 1% BSA in PBS overnight at 4°C. Samples were then incubated with Hoechst 33258 at 10 µg/ml in PBT for 1 hr at RT, washed 3x with PBT for 5 min each, and then washed into PBS at 4°C overnight. PBS was replaced with 95% glycerol with 2.5 mg/ml DABCO, and the larvae were mounted for microscopy. In peptide-blocked controls, the corresponding immunogenic peptides dissolved in PBS or DMSO at 0.5 mg/ml – 1 mg/ml were added to diluted primary antibodies (XM_021052412 at 1:200 [1.5 µg/ml], XM_021052404 at 1:750 [0.6 µg/ml], and XM_021052381 at 1:200 [2 µg/ml]) and rotated at 1 hr at RT before being added to samples, which were then processed as described.

## U18666A exposure in *Aiptasia* and *A. digitifera*

Symbiotic and aposymbiotic *Aiptasia* polyps were allowed to attach for 2 d in 6-well culture plates before exposure to U18666A (U3633, Sigma Aldrich) in DMSO at the indicated concentrations in FASW; final percentage of DMSO was <0.05%. Polyps were cultured at 26°C at 12:12 L:D and photographed daily, followed by wash and drug re-addition. Symbiont density per anemone was quantified by homogenization in 200 µl ultrapure water with 0.01% SDS using a 23G needle and 1 ml syringe, after which samples were quantified for cells by visual particle counter (TC20, BioRad) and for total protein by the Pierce BCA Protein Assay Kit (23227, Thermo Fisher Scientific). *Acropora* polyps hosting *Symbiodiniaceae* SSB01 were exposed to U18666A as described, except that they were cultured in FNSW.

## Microscopy

Confocal microscopy of NPC2 immunofluorescence was performed using a Leica SP8 system with an HC PL Apo CS2 63x/1.30 GLYC objective. Hoechst was excited at 405 nm and detected at 410–501 nm, and algal autofluorescence was excited at 633 nm and detected at 645–741 nm. In a second sequential scan, Alexa-488 (secondary antibody) was excited at 496 nm and detected at 501–541 nm. Z-stacks were collected with a step size of 0.5 µm and 3x line averaging. A zoom factor of 5 or,

for whole larvae, 1.33, was used, and a pinhole of 1 Airy unit. Quantification and imaging NPC2 IF over a time-course was carried out using a Nikon Eclipse Ti epifluorescence compound microscope with a Plan Apo λ 40x objective, Sola light source, and GFP filter set. Images were captured with a Nikon DS-Qi2 with an exposure time of 1 s. Fluorescence microscopy of *Aiptasia* adults was carried out using a Nikon SMZ18 fluorescence stereoscope with a 0.5X objective; endogenous autofluorescence of symbiont photosynthetic antennae was visualized with a Texas Red filter set, and images were captured at magnification 15X with an Orca-Flash4.0 camera (C11440, Hamamatsu) at 300 ms exposure using Nikon Elements software and processed in Fiji (*Schindelin et al., 2012*). *Acropora* polyps were photographed as described (*Wolfowicz et al., 2016*), and fluorescence was quantified in Fiji (*Schindelin et al., 2012*) as total fluorescence in the polyp area minus adjacent background.

## Statistical information

In GC/MS-based sterol profiling, shown in *Figure 1A* are representatives of n = 3 (*Aiptasia,* SSB01) or n = 2 (*Acropora,* SSA01, CCMP2556) samples each, shown in *Figure 1B* are representatives of n = 3 samples each, and shown in *Figure 1C* and *Figure 1—figure supplement 2* are averages (*A*) and representatives (*B*) of n = 2 samples each. In gene expression analyses by qPCR (*Figure 3A*), shown are average values of 6 samples per condition, six animals per sample, each in technical duplicate for *Aiptasia*. For *Nematostella* (*Figure 3—figure supplement 2*), shown are average values of 2 animals per sample, two samples per condition, each in technical duplicate. For *Acropora* (*Figure 3— figure supplement 2*), shown are averages of two biological replicates, each in technical duplicate. In NPC2 immunofluorescence in *Aiptasia* larvae (*Figure 3F*), shown is a representative of two independent experiments, each with triplicate samples of >50 larvae per time-point. In sterol-blocking U18666A pharmacological experiments, shown are representative images of n = 3 polyps per anemone type and condition, with all anemones shown in *Figure 3—figure supplement 6*; symbiotic representatives are from one of three replicate experiments (*Figure 3G + H*). Quantification of symbiont density (*Figure 3H*) in n = 3 anemones per condition, each in technical duplicate. Shown in *Figure 3—figure supplement 7* are representative images of n = 5 polyps across duplicate wells (n = 4 for 10 µM). In immunoprecipitation-lipidomics experiments, shown are averages of duplicate samples, with representative experiments shown of two (*Figure 4C*) or three replicate experiments (*Figure 4A + B*). For experiments assessing soluble NPC2 at different pHs, ratios of ID at pH 7 divided by that at pH 5 were calculated from duplicate-loaded bands per protein per pH condition from a single blot, from three (canonical NPC2 XM_021041171; non-canonical NPC2 XM_021052404) or six (non-canonical NPC2 XM_021052412; crmCherry alone) replicate experiments (*Figure 4F*). Shown in *Figure 4E* are one pair of treatments in a representative experiment, from the aforementioned number of replicate experiments.

## Acknowledgements

We thank Britta Brügger, Christian Lüchtenborg, and Iris Leibrecht for assistance with sterol-binding lipidomics; Masayuki Hatta for *Acropora* coral and embryo collection and advice; Thomas Holstein for *Nematostella* provision, qPCR instrument access, and together with Sergio P Acebrón for cell culture materials; Natascha Bechtoldt for qPCR technical assistance; Emmanuel Gaquerel, Anne Terhalle, Michael Büttner, and Gernot Poschet (COS Metabolomics Core Technology Platform [MCTP]) for GC/MS instrument access and advice; Frauke Graeter, Agnieszka Obarska-Kosinska, Rebecca Wade, and Aura Navarro Quezada for input on theoretical and evolutionary analyses. We thank Life Science Editors for editorial assistance with the manuscript. Support was provided to EAH, VASJ, IM, and AG by the Deutsche Forschungsgemeinschaft (DFG) (Emmy Noether Program Grant GU 1128/3–1), the European Commission Seventh Framework Marie-Curie Actions (FP7-PEOPLE-2013-CIG), the H2020 European Research Council (ERC Consolidator Grant 724715), the Boehringer Ingelheim Stiftung (Exploration Grant), and the future concept of Heidelberg University within the Excellence Initiative by German federal and state governments to AG; to VASJ by the European Molecular Biology Organization (Long-Term Fellowship); and to DK and TS by CellNetworks Heidelberg and the German Research Foundation (DFG, TRR83 and SFB1324) to Britta Brügger.

# Additional information

## Funding

| Funder | Grant reference number | Author |
| --- | --- | --- |
| Deutsche Forschungsgemeinschaft | GU 1128/3-1 | Annika Guse |
| European Commission | FP7-PEOPLE-2013-CIG | Annika Guse |
| Horizon 2020 Framework Programme | 724715 | Annika Guse |
| Boehringer Ingelheim Stiftung | Exploration Grant | Annika Guse |
| European Molecular Biology Organization | LTF | Victor Arnold Shivas Jones |
| Universität Heidelberg | Excellence Initiative | Annika Guse |
| Deutsche Forschungsgemeinschaft | TRR83 | David Kvaskoff Timo Sachsenheimer |
| Deutsche Forschungsgemeinschaft | SFB1324 | David Kvaskoff Timo Sachsenheimer |

The funders had no role in study design, data collection and interpretation, or the decision to submit the work for publication.

## Author contributions

Elizabeth Ann Hambleton, Conceptualization, Formal analysis, Supervision, Investigation, Visualization, Methodology, Writing—original draft, Project administration, Writing—review and editing, Directed the research, Primarily conducted the experiments and analyses with contributions from IM with immunoprecipitations, pH experiments, cloning, symbiont genotyping, and Western blots, Conducted and analysed sterol-binding lipidomics with DK and TS; Victor Arnold Shivas Jones, Formal analysis, Investigation, Visualization, Methodology, Writing—review and editing, Conducted the immunofluorescence, confocal imaging, and IF quantification; Ira Maegele, Investigation, Visualization, Methodology, Writing—review and editing, Contributed towards conducting the experiments and analyses with immunoprecipitations, pH experiments, cloning, symbiont genotyping, and Western blots; David Kvaskoff, Data curation, Investigation, Methodology, Conducted and analysed sterol-binding lipidomics with EAH and TS; Timo Sachsenheimer, Investigation, Conducted and analysed sterol-binding lipidomics with DK and EAH; Annika Guse, Conceptualization, Resources, Formal analysis, Supervision, Funding acquisition, Visualization, Methodology, Writing—original draft, Project administration, Writing—review and editing, Directed the research

## Author ORCIDs

Elizabeth Ann Hambleton (iD) https://orcid.org/0000-0003-0204-5037
Victor Arnold Shivas Jones (iD) https://orcid.org/0000-0002-8712-3454
Ira Maegele (iD) https://orcid.org/0000-0002-2324-3561
David Kvaskoff (iD) https://orcid.org/0000-0002-5569-7226
Annika Guse (iD) https://orcid.org/0000-0002-8737-9206

## Decision letter and Author response

Decision letter https://doi.org/10.7554/eLife.43923.024
Author response https://doi.org/10.7554/eLife.43923.025

# Additional files

## Supplementary files

• Supplementary file 1. Newly annotated or corrected NPC2 homologues.
DOI: https://doi.org/10.7554/eLife.43923.019

• Supplementary file 2. Genome, transcriptome, and proteome accession information.
DOI: https://doi.org/10.7554/eLife.43923.020

• Supplementary file 3. Primers for qPCR in *Aiptasia*, *Nematostella*, and *Acropora digitifera*.
DOI: https://doi.org/10.7554/eLife.43923.021

• Transparent reporting form
DOI: https://doi.org/10.7554/eLife.43923.022

## Data availability

All data generated or analysed during this study are included in the manuscript and supporting files.

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
