## [Decision Letter]

Thank you for submitting your article "Sterol transfer by atypical cholesterol-binding NPC2 proteins in coral-algal symbiosis" for consideration by *eLife*. Your article has been reviewed by two peer reviewers, and the evaluation has been overseen by a Reviewing Editor and Ian Baldwin as the Senior Editor. The reviewers have opted to remain anonymous.

The reviewers have discussed the reviews with one another and the Reviewing Editor has drafted this decision to help you prepare a revised submission.

Both reviewers have similar concerns on some of the experimental results, especially with respect to U18666A and its specificity. One reviewer specifically asked for a control experiment with aposymbiotic anemones to verify the effects of the drug. Normally, if reviewers request further experiments, it is grounds for rejecting a paper, however, based on their positive comments in general, we are asking that you perform these experiments and submit a revised manuscript that addresses these and the other concerns raised. Both reviewers felt that these additional experiments could be accomplished in less than two months. In some cases, the comments can be addressed by adjusting the writing of the manuscript.

*Reviewer #2:*

This study is extremely interesting, novel, and well conducted. It investigated the role of NPC2 proteins in transporting sterols in cnidarian-algal symbioses. As cnidarians are sterol-auxotrophs, they can only get sterols from their diet or symbiotic partners. The level of detail and variety of techniques that were used is remarkable, and even more so for a study on cnidarians with relevance for coral reefs.

I have identified a few issues (and also many interesting questions that are not problematic), and I hope the authors can address at least for some of them with additional experiments.

1) The experiments with the drug U18666A are problematic. This drug induced loss of symbionts from the host anemones, tissue loss, and shortening of tentacles, which the authors interpreted as direct evidence for disruption of NPC1/2-mediated sterol transport from symbionts to host, and especially from non-canonical NPC2s (which are more highly expressed during symbiosis). However, the effects seen suggest toxicity, possibly due to unspecific effects of the high doses used together with general inhibition of sterol export throughout cnidarian tissues (and *Symbiodinium*, if they have NPCs).

Although NPC2s are certainly present in symbiosome-containing cells, the drug inhibits NPC1 which surely is present throughout the anemone tissues.

These issues must be addressed by looking at the effects of U18666A on aposymbiotic anemones, and ideally also by measuring sterol composition in symbiotic anemones treated with U18666A (which should display reduce abundance of *Symbiodinium*-derived lipids).

2) It seems that the antibodies generated in this study specifically recognize three non-canonical NPC2s. Then, what antibodies were used to detect the canonical NPC2 (XM_021041171) by Western blot in Figure 4E? (if they were the same antibodies, it would question the specificity of the antibodies and interpretation of results).

General question about antibody validation: Were peptide-preabsorption controls performed for Westerns and microsopy? (If not, they must be done).

Do these antibodies recognize NPCs from *Acropora*?

3) The sterol composition of cnidarian tissues (which contain symbionts) was compared to that from cultured Symbiodinium. The authors have cleverly addressed potential contamination of cnidarian tissues with symbiont sterols by examining symbiont-free eggs, which receive sterols from their parent. However, while symbiotic anemones contain some sterols that are present in *Symbiodinium* and absent in aposymbiotic anemones, they also lack many of the most abundant sterols present in *Symbiodinium* (e.g. stigmasterol/gorgosterol-like, and ergost-tetrol-like). The authors explain these results by differential sterol transport and accumulation; however, is it also possible that *Symbiodinium* produces different sterols while in symbiosis, and in fact it is well established that many physiological aspects of free-living (or cultured) *Symbiodinium* is different from symbiotic *Symbiodinium*. This could be easily examined by separating symbionts from whole cnidarian tissues by a slow-speed centrifugation step and comparing the lipid composition of the pelleted symbionts to the host tissue that remains in the supernatant. This technique should also be used for the experiment suggested in point (1) above. (Note: this is not an essential experiment for this paper, but it could provide very interesting information).

*Reviewer #3:*

The manuscript by Guse and colleagues presents several interesting new findings related to sterol biology in cnidarian corals and anemones, without or with their endosymbionts. In agreement with prior literature they demonstrate that sterols transfer from the symbiont to the host, and further show the time course of the changing sterol composition, which varies with the degree of symbiosis. Also in agreement with a previous report, they show NPC2 localization to the symbiont membrane. In addition they compare so-called canonical NPC2 proteins with non-canonical NPC2 proteins, demonstrating several differences including a greater degree of upregulation of the non-canonical NPC2's during symbiosis (i.e. between fasted and fed cnidarians), a primary sequence difference which indicates greater acid stability for the non-canonical NPC2's, and possibly somewhat greater acid tolerance compared with the canonical NPC2's. They further use the U18666A compound, known to directly bind and inhibit the function of mammalian NPC1, and show that it causes severe morphological derangement in host animals.

While the work appears quite well done and the results are interesting, the conclusions seem to outpace the actual data.

1) There is little selectivity observed in the transfer of different sterol species between symbiont and host, yet the summarizing phrases used are that a 'unique sterol mix' is transferred to the host. I suppose you can say that the sterol composition of a dinoflagellate is different and therefore unique compared to a fasted anemone, but the summation implies selectivity and this is clearly not the case. (In fact this is in contrast to mammalian sterol transfer; for example in humans only cholesterol is transferred from the gut into the intestinal cell, while minimally different plant sterols are excluded).

2) The case for greater acid stability of the non-canonical compared to canonical NPC2's is based on the presence of one GH signature sequence in non-canonical but not NPC2's, and the purported smaller decrease in NPC2 protein content under pH5 vs. pH7 for non-canonical NPC2's than canonical NPC2's. But examination of Figures 4E and 4F shows only one non-canonical protein and two canonical NPC2's, one of which did not differ in apparent acid stability compared to the non-canonical NPC2. A direct test of several purified proteins from each class would be needed to support this conclusion. (In this regard, the mammalian NPC2 protein was shown to have higher sterol transfer activity at acidic pH than at neutral pH (Cheruku et al., 2006 J Biol Chem); functional analysis, either of sterol binding or perhaps transfer, as a function of pH would strengthen this aspect of the invesitgators' story).

3) The U18666A experiments show marked derangement of the cnidarian host by the compound (Figure 3G and 3H). The "U-compound" has been demonstrated to target NPC1 (again, in mammalian cells), as shown in Vance, 2010. This experiment, therefore, does not shed light on the NPC2 proteins, but rather leads to the general statement that sterols are important in tissue homeostasis (main text, sixth paragraph)., which is not surprising. It would be useful to have more information on NPC1 proteins in these organisms, e.g. is a similar upregulation seen upon initiation of symbiosis? As it stands, the U compound experiment does not provide much information.

4) The conclusion that the sterol harvesting machinery (NPC2 and NPC1, presumably) are "key to symbiosis" (Abstract) is not well supported, as no experiment directly addressed this point, rather the results are generally correlative.

Overall this is a well done study with several interesting confirmatory and novel findings, but the major conclusions drawn do not seem sufficiently well supported by the results.

---

## [Author Response]

Reviewer #2:[…] I have identified a few issues (and also many interesting questions that are not problematic), and I hope the authors can address at least for some of them with additional experiments.1) The experiments with the drug U18666A are problematic. This drug induced loss of symbionts from the host anemones, tissue loss, and shortening of tentacles, which the authors interpreted as direct evidence for disruption of NPC1/2-mediated sterol transport from symbionts to host, and especially from non-canonical NPC2s (which are more highly expressed during symbiosis). However, the effects seen suggest toxicity, possibly due to unspecific effects of the high doses used together with general inhibition of sterol export throughout cnidarian tissues (and Symbiodinium, if they have NPCs).Although NPC2s are certainly present in symbiosome-containing cells, the drug inhibits NPC1 which surely is present throughout the anemone tissues.These issues must be addressed by looking at the effects of U18666A on aposymbiotic anemones, and ideally also by measuring sterol composition in symbiotic anemones treated with U18666A (which should display reduce abundance of Symbiodinium-derived lipids).

We are indeed fully aware that the experiments with U18666A are somewhat problematic because, as the reviewer points out, the drug does not specifically target NPC2 in the symbiosomes, but rather NPC1 and thus sterol egress from the lysosome in *all* cells throughout the anemone, independent of whether such cells contain symbionts. Because of this profound effect on animal physiology, again as the reviewer rightfully points out, some toxicity effects are to be expected. However, the question remains of whether symbiotic animals (which presumably have, due to the symbionts, higher levels of active sterol transfer when compared to non-symbiotic anemones) respond more drastically to this drug and whether this effect can be distinguished from the drug´s overall effects.

To specifically address this, we have now repeated and extended the experiment according to the reviewer’s suggestion by including more time-points, as well as comparable aposymbiotic anemones as controls (for more details, see the updated Materials and methods, subsections “U18666A exposure in *Aiptasia* and *A. digitifera*” and “Statistical information”). Please also note that for the new experiments, we used two U18666A concentrations (0.2 and 0.4 µM), which differ from the previously used ones (0.2, 1 and 2 µM). The reason for reducing the concentration is that this time we used smaller animals and in a preceding pilot experiment, we found 0.2 and 0.4 µM to be suitable to maintain anemones this size for 10-14 days in a state not too physically detrimental.

We find that indeed at both concentrations tested, symbiotic anemones show more rapidly tentacle retraction and shrinkage than their aposymbiotic counterparts (new Figure 3G and new Figure 3—figure supplement 6). Accordingly, quantification of symbiont density as number of symbionts per µg host tissue suggests that over a time-period of 10 days, symbionts get lost in a dose-dependent manner (new Figure 3H). (please note that quantification of symbiont density is somewhat difficult because of the natural variability in symbiont densities in anemones). Together, this suggests that inhibition of sterol egress from lysosomes impairs symbiosis stability and thus may lead to symbiont loss. Understanding the mechanistic details of this phenomenon will be an important endeavor for the future. We hope that we have addressed the reviewer´s concern with this additional experiment.

We have further clarified the manuscript text to reflect this assessment (please see Results and Discussion, sixth paragraph). In addition to the updated Figure 3 and new Figure 3—figure supplement 6, we have moved the accompanying coral experiments to Figure 3—figure supplement 7 and updated the corresponding figure legends.

Finally, we would also like to mention that while we thought the suggestion raised is an interesting idea, we did not attempt to measure whether the sterol composition in the host tissue of symbiotic anemones changes upon drug addition, mostly because we feel that measurable changes in sterol proportions may be technically difficult to determine. Our GC-MS analysis here is not per se quantitative and is not sensitive enough to pick up small quantitative changes in sterol abundance, hence our use for comparative abundances. To address this interesting point, we would need to precisely develop new and more sensitive and highly quantitative assays.

2) It seems that the antibodies generated in this study specifically recognize three non-canonical NPC2s. Then, what antibodies were used to detect the canonical NPC2 (XM_021041171) by Western blot in Figure 4E? (if they were the same antibodies, it would question the specificity of the antibodies and interpretation of results).

Indeed we generated antibodies that specifically recognize the three endogenous, non-canonical NPC2s via Western blot using *Aiptasia* extracts (Figure 3B and new Figure 3—figure supplement 1). However, for Figure 4E, we express *Aiptasia* NPC2s recombinantly in mammalian cell culture, and the detection of all NPC2 proteins in Figure 4E uses an anti-mCherry antibody that recognizes the same C-terminal mCherry-tag on every expressed NPC2 (Materials and methods, subsection “Immunoprecipitation-lipidomics of NPC2-sterol binding”). To clarify, we have now added this information in the legend for Figure 4E.

General question about antibody validation: Were peptide-preabsorption controls performed for Westerns and microsopy? (If not, they must be done).

To address this point, we have now included a new supplementary figure entirely dedicated to validating our distinct NPC2 antibodies (new Figure 3—figure supplement 1). The protein sequences of the non-canonical NPC2s are relatively conserved, and we therefore chose peptide antigens based on the C-terminal regions, which substantially differ. We included an alignment of the NPC2 proteins and indicated the regions that were chosen as antigens aiming for maximal sequence diversity (A). Next, we used dot blot analysis to show that the antibodies do not show any cross-reactivity for the distinct peptides, supporting their specificity for individual NPC2 proteins (B). We further validated this using whole anemone extract and Western blotting as well as using IF: we find that peptide pre-absorption blocks NPC2 detection effectively for all three antibodies in both methods (C+D).

We have added a reference to this new figure supplement in the manuscript (Results and Discussion, fifth paragraph) along with the new corresponding figure legend, and we have added the corresponding information to the Materials and methods section.

Do these antibodies recognize NPCs from Acropora?

We have not tried this experimentally but we do not expect that our antibodies would recognize *Acropora* NPC2 because of their sequence diversity. Specifically, none of the antibody epitopes have more than 3 of 15 residues in common with any *Acropora* NPC2s, making interspecies cross-reactivity unlikely.

3) The sterol composition of cnidarian tissues (which contain symbionts) was compared to that from cultured Symbiodinium. The authors have cleverly addressed potential contamination of cnidarian tissues with symbiont sterols by examining symbiont-free eggs, which receive sterols from their parent. However, while symbiotic anemones contain some sterols that are present in Symbiodinum and absent in aposymbiotic anemones, they also lack many of the most abundant sterols present in Symbiodinium (e.g. stigmasterol/gorgosterol-like, and ergost-tetrol-like). The authors explain these results by differential sterol transport and accumulation; however, is it also possible that Symbiodinium produces different sterols while in symbiosis, and in fact it is well established that many physiological aspects of free-living (or cultured) Symbiodinium is different from symbiotic Symbiodinium. This could be easily examined by separating symbionts from whole cnidarian tissues by a slow-speed centrifugation step and comparing the lipid composition of the pelleted symbionts to the host tissue that remains in the supernatant. This technique should also be used for the experiment suggested in point (1) above. (Note: this is not an essential experiment for this paper, but it could provide very interesting information).

We fully agree with the reviewer: it is entirely possible that *Symbiodinium* produces distinct mixes of sterol depending on its life style: free-living or as a symbiont inside the host. To specifically test this possibility, we have performed the experiment as suggested by the reviewer, wherein symbiotic anemone homogenates were separated by centrifugation and the symbiont-containing pellets were compared to symbionts in culture via our GC/MS sterol profiling analysis (for the updated Materials and methods). Please note that we attempted pilot experiments to test the best method for such centrifugal separation, with the best parameters then used for the new experiment now in the manuscript. However, the symbiont-containing pellets still did contain a substantial amount of anemone host tissue, which we did our best to quantify in both number and% contribution to each sample (new Figure 1—figure supplement 2A).

Notably, we found evidence in these experiments to indicate that indeed, the symbionts may be changing the abundance/synthesis of their sterols while in residence in anemones. As the reviewer points out, the ergost-tetrol-like sterol (light purple) is abundant in free-living symbionts in culture but not found (no peak in the GC/MS data, sterol contribution 0%) in the symbiont-containing pellets (new Figure 1C and new Figure 1—figure supplement 2B). We reasoned that if the symbionts were not changing their sterol profiles from free-living to symbiotic, this sterol would almost certainly be detected in the symbiont-containing pellets especially considering its relative abundance to stigmasterol/gorgosterol-like (dark purple), which is present in the symbiont-containing pellets (albeit also in a very small proportion that could suggest that it too may change upon symbiosis). Overall, the technical limitations of such a separation approach prevent us from drawing too many specific conclusions as to exactly which sterols and precisely to what extent are differentially present/synthesized in symbionts in culture vs. in residence. However, we conclude that these experiments demonstrate that this occurs to at least some extent.

Because of these interesting additions to the GC/MS data, we have amended the manuscript text to include this new experiment (Results and Discussion, third paragraph) in addition to the above-mentioned addition to the Materials and methods section, the accompanying new Figure 1C and Figure 1—figure supplement 2, and the new corresponding figure legends. Of course, we welcome any additional comments or discussion from the reviewers and editors and, seeing as how this new experiment accompanies new text, we are open to further edits on these new portions.

Reviewer #3:The manuscript by Guse and colleagues presents several interesting new findings related to sterol biology in cnidarian corals and anemones, without or with their endosymbionts. In agreement with prior literature they demonstrate that sterols transfer from the symbiont to the host, and further show the time course of the changing sterol composition, which varies with the degree of symbiosis. Also in agreement with a previous report, they show NPC2 localization to the symbiont membrane. In addition they compare so-called canonical NPC2 proteins with non-canonical NPC2 proteins, demonstrating several differences including a greater degree of upregulation of the non-canonical NPC2's during symbiosis (i.e. between fasted and fed cnidarians), a primary sequence difference which indicates greater acid stability for the non-canonical NPC2's, and possibly somewhat greater acid tolerance compared with the canonical NPC2's. They further use the U18666A compound, known to directly bind and inhibit the function of mammalian NPC1, and show that it causes severe morphological derangement in host animals.While the work appears quite well done and the results are interesting, the conclusions seem to outpace the actual data.

We thank the reviewer for the assessment and very helpful comments. We now aimed to better match our conclusions with the actual data to address the reviewer´s overall concern.

1) There is little selectivity observed in the transfer of different sterol species between symbiont and host, yet the summarizing phrases used are that a 'unique sterol mix' is transferred to the host. I suppose you can say that the sterol composition of a dinoflagellate is different and therefore unique compared to a fasted anemone, but the summation implies selectivity and this is clearly not the case. (In fact this is in contrast to mammalian sterol transfer; for example in humans only cholesterol is transferred from the gut into the intestinal cell, while minimally different plant sterols are excluded).

We agree with the reviewer´s assessment that the phrase ‘a unique sterol mix is transferred’ may be misleading when compared to the situation described above for intestinal cells in mammals. However, as the reviewer points out, the sterol composition in free-living symbionts differs from the sterol mix recovered from anemones and their eggs. This difference may have various reasons: the symbiont only transfers a subset, the host specifically accumulates certain sterols, the symbiont changes the composition of its produced and thus transferred sterols upon entering symbiosis. Alternatively, the host may convert transferred sterols into the most suitable for the host. However, the latter seem unlikely to be the dominant mode because our mix and match experiments demonstrate that the host composition is a direct response of the housed symbiont and thus differs for distinct symbionts.

We conclude that the host has the ability to flexibly use distinct sterol mixes, thereby adapting to their symbionts. Further, our results show that corals appear to be capable of living basically without cholesterol, further supporting the flexibility and adaptability of symbiotic cnidarians, especially in comparison with e.g. mammals that selectively take up cholesterol and reject minimally different plant sterols, as the reviewer mentions above.

To date we can only speculate about the underlying mechanism, and we agree with the reviewer that we should avoid phrases such as “a unique sterol mix is transferred” because this implies an active, selective mechanism for which we currently don’t have sufficient data. We have therefore now changed the manuscript and used more open and thus hopefully appropriate phrases without diminishing our findings (please see Abstract and Results and Discussion, last paragraph).

2) The case for greater acid stability of the non-canonical compared to canonical NPC2's is based on the presence of one GH signature sequence in non-canonical but not NPC2's, and the purported smaller decrease in NPC2 protein content under pH5 vs. pH7 for non-canonical NPC2's than canonical NPC2's. But examination of Figures 4E and 4F shows only one non-canonical protein and two canonical NPC2's, one of which did not differ in apparent acid stability compared to the non-canonical NPC2. A direct test of several purified proteins from each class would be needed to support this conclusion. (In this regard, the mammalian NPC2 protein was shown to have higher sterol transfer activity at acidic pH than at neutral pH (Cheruku et al., 2006 J Biol Chem); functional analysis, either of sterol binding or perhaps transfer, as a function of pH would strengthen this aspect of the invesitgators' story).

We agree with the reviewer´s critical assessment. A more comprehensive analysis is required to determine the specific biological function(s) of the non-canonical NPC2 in symbiosis when compared to canonical NPC2 and among each other. Nevertheless, we find that *two* non-canonical NPC2s show a higher acid-tolerance when compared to *one* canonical NPC2. The result is only significant for one of the proteins tested, but the trend holds for both (Figure 4F). The tested canonical NPC2 is by far the most abundant among its canonical counterparts in the organism (Figure 3A), and the two tested non-canonical NPC2s are the highest up-regulated of their counterparts in symbiosis (Figure 3A) suggesting that these proteins play important roles. However, we agree that purified proteins would indeed be ideal to fully test all canonical and non-canonical NPC2s directly. We are indeed working towards that goal; however, it has proven difficult so far to express the proteins of interest in bacteria and/or insect cells after multiple attempts in both systems, and the yield from expression in mammalian cells was too low to establish fully-controlled in vitro biochemical assays testing for protein stability, sterol binding and transfer activity.

To our knowledge and after consultation with colleagues in molecular dynamics and modeling, pH stability of proteins is difficult to predict based on pure sequence analysis and always has to be combined with experimental data. The presence of additional Histidines may be an indication for higher stability at low pH but is for sure not a proof.

It was neither our goal to imply that enhanced stability under low pH is the *only* potential function for the non-canonical NPC2s nor that the identified GH-motif is the one and only reason for our experimental findings. Therefore, and in agreement with the reviewer´s criticism, we have now changed the text in the manuscript to avoid over-simplification and generalization on this topic (please see Abstract, Results and Discussion, last paragraph).

3) The U18666A experiments show marked derangement of the cnidarian host by the compound (Figure 3G and 3H). The "U-compound" has been demonstrated to target NPC1 (again, in mammalian cells), as shown in Vance, 2010. This experiment, therefore, does not shed light on the NPC2 proteins, but rather leads to the general statement that sterols are important in tissue homeostasis (main text, sixth paragraph)., which is not surprising. It would be useful to have more information on NPC1 proteins in these organisms, e.g. is a similar upregulation seen upon initiation of symbiosis? As it stands, the U compound experiment does not provide much information.

We fully agree with the reviewer that the U18666A experiment does not shed any light on the function of NPC2 proteins specifically, but rather confirms an indeed not-too-surprising effect on general tissue homeostasis due to impaired sterol transfer in all cells throughout the organism. Indeed we found a single NPC1 gene in Aiptasia that is overall highly expressed but not upregulated in symbiosis. However, it may be speculated that symbiosomes are hotspots of sterol transfer where sterols are transported from the symbiont via the symbiosome lumen and NPC1 into the cytosol are more drastically affected by the addition of U18666A than lysosomes in cells that do not contain symbionts. For example, blocked egress of symbiont-derived sterols from the symbiosome lumen may lead to an accumulation of sterols within the symbiosome. This accumulation may be unhealthy for the cell which may in turn lead to symbiont loss (similar to a lysosomal storage disease phenotype in mammals).

To address these concerns by both reviewers, we repeated the U18666A experiment using both symbiotic and aposymbiotic anemones in order to determine whether any differences in response could relate to symbiosis. In line with this hypothesis, we now find that symbiotic anemones are indeed more severely affected by U18666A than their aposymbiotic counterparts (new Figure 3G and H and Figure 3—figure supplement 6) suggesting that inhibiting sterol egress via NPC1 affects symbiosis stability.

This being said, we fully agree with the reviewer´s opinion, and we have changed the manuscript text to avoid implying insights into the function of non-canonical NPC2 proteins but rather clearly state the potential caveats of using this drug. Please see the explanation we provide above in our response to point #1 from reviewer #2. Our changes to the main text are found in the sixth paragraph of the Results and Discussion. In addition to the updated Figure 3 and new Figure 3—figure supplement 6, we have moved the accompanying coral experiments to Figure 3—figure supplement 7 and updated figure legends and Materials and methods sections (subsections “U18666A exposure in *Aiptasia* and *A. digitifera*” and “Statistical information”).

4) The conclusion that the sterol harvesting machinery (NPC2 and NPC1, presumably) are "key to symbiosis" (Abstract) is not well supported, as no experiment directly addressed this point, rather the results are generally correlative.

We agree with the reviewer and changed the specific phrase, as well as a few other phrases that may be misleading, in the Abstract.

Overall this is a well done study with several interesting confirmatory and novel findings, but the major conclusions drawn do not seem sufficiently well supported by the results.

Again, we thank the reviewer for the overall positive assessment and hope that we have addressed the reviewer´s concerns by changing the manuscript in a satisfying way.